# CAN LONG-CONTEXT LANGUAGE MODELS UNDERSTAND LONG CONTEXTS?

## ABSTRACT

Large language models (LLMs) have garnered significant attention due to their remarkable performance across various NLP tasks. However, the fixed context window length of the transformer architecture renders them incapable of memorizing and understanding extremely long inputs. There are tremendous works in designing effective and advanced techniques to enlarge LLMs' context window size, which call for high demands on developing high-quality benchmark datasets to evaluate LLMs' long context understanding ability. There are some existing datasets for this purpose. However, they face the problems of (1) shorter text length compared to modern LLMs' context window length, (2) out-of-date documents that may already been part of the the training corpus of modern LLMs, and (3) most of the tasks involve short dependency tasks—there are few questions that really need LLMs to collect information across the entire document (which we refer to as *long dependency tasks*). In this paper, we present **LooGLE**, a **Lo**ng **Co**ntext **G**eneric **L**anguage **E**valuation benchmark for LLM long context understanding. It contains up-to-date documents (all from after 2022), with over 24k tokens on average, and over 6k newly generated questions from diverse domains and categories. Specifically, we recruited a group of human labelers to read 140 long documents in our benchmark, and asked them to compose about 1.1k QA pairs satisfying our long dependency requirements. These 1.1k high-quality QA pairs are each constructed through a 3-step cross-validation process by 2 annotators, aiming to provide the most accurate evaluation of LLMs' ability on long dependency questions currently available. Upon a comprehensive evaluation of 8 state-of-the-art LLMs on **LooGLE**, we find that: (1) Commercial models generally outperform open-sourced models. (2) LLMs are more skilled at short dependency tasks like short QA and cloze but still struggle on performing real long dependency tasks. (3) In-context learning and chain of thoughts only bring incremental improvement for long context understanding.(4) Retrieval-based techniques significantly contribute to improvement on short QA whereas many techniques for extending the context window length through optimized transformer architecture or positional encoding can hardly resolve long context understanding. As such, **LooGLE** not only provides a systematic and comprehensive evaluation schema on long-context LLMs, but also sheds light on future development of enhanced models towards "true long-context understanding".

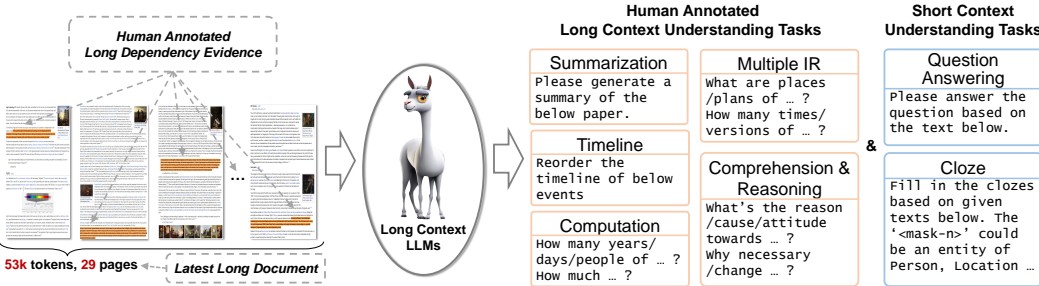

Figure 1: Our LooGLE benchmark for long context understanding

# 1 INTRODUCTION

The pursuit of enabling large language models (LLMs), such as ChatGPT (Brown et al., 2020; OpenAI, 2023), to go beyond their limited context window size so as to process, comprehend, or even learn from long-context textual information (Ding et al., 2023; Dao et al., 2022; Chi et al., 2023; Bulatov et al., 2023) is inevitable for next generation of language intelligence attributed to its wide applications on real-world scenarios, such as domain-specific knowledge understanding, long-context conversational generation, long story or code generation, *etc*.

Meanwhile, there is an increasing need for high-quality benchmarks with much longer text lengths and more challenging tasks to provide comprehensive evaluations. However, traditional benchmarks (Cohan et al., 2018; Sharma et al., 2019; Huang et al., 2021) often fall short in text length with an average number of thousands of words (s Koˇciský et al., 2018; Yang et al., 2018). Besides, existing benchmarks automatically collect possibly outdated documents from existing datasets (Shaham et al., 2022; Trivedi et al., 2022; Wang et al., 2022; Angelidis et al., 2020), which might lead to data leakage in pretrained LLMs and make the evaluation inaccurate. Furthermore, the long texts are often restricted to domain-specific articles, making it challenging to evaluate LLMs' ability on generic tasks and domains. Finally, it is important to note that tasks in existing benchmarks are primarily **short dependency** tasks. These tasks only require LLMs to retrieve answers from one specific sentence or paragraph, without truly testing LLMs' ability to collect pieces of information from paragraphs across the entire document and summarize them into an answer, which we call **long dependency** tasks.

To mitigate the shortcomings of existing datasets, in this paper, we introduce a novel benchmark **LooGLE** , abbreviated for **Lo**ng **C**ontext **G**eneric **L**anguage **E**valuation, to evaluate the long context understanding abilities of LLMs, as illustrated in Fig. 1. Our benchmark has the following advantages:

- **Extra-long realistic documents**. It contains 776 latest gathered and extremely long documents with an average of 19.3k words. Based on the documents, we further generate 6,448 questions without distribution bias for a more generalized assessment. On one hand, they can better evaluate LLMs' capability on memorizing and understanding longer text that is far beyond their context window size. On the other hand, the excessive length is well-suited to the common usage of long text scenarios.
- **Manually designed both short and long dependency tasks.** It is composed of 7 major tasks with a total of 6,448 questions to evaluate LLMs' ability to understand both short and long dependency content. We refer to "long dependency" tasks as those that require an understanding of the inter-dependency across multiple pieces of evidence widely spanning over the entire long text. We delicately designed 5 types of long dependency tasks and recruited a group of human labelers to manually create 1,101 long dependency Question-Answer (QA) instances, despite the high costs and huge effort involved in this process.
- **Up-to-date documents.** Our benchmark comprises texts all published after 2022 ensuring that modern LLMs such as GPT-3.5 (gpt-3.5-turbo-0603) and GPT-4 have not been pretrained on these documents, **forcing them to rely on their in-context learning ability** rather than memorization. In contrast, existing benchmarks contain out-of-date content, whose world knowledge may have already been learned by LLMs and thus is less convincing to assess LLMs' true long context understanding ability.
- **Cross-domain generic data.** Our benchmark is derived from popular open-source documents, including arXiv papers, Wikipedia articles, and movie and TV scripts, spanning diverse domains and multiple categories such as academia, history, sports, politics, arts, events, and entertainment.

Table 1: Comparison with other long-context benchmarks

| Dataset | Avg. Words | # of Docs. | # of Ques. | Manually Label | Long Dependency Tasks | | | | |
|---|---|---|---|---|---|---|---|---|---|
| | | | | | SUMM | IR | TR | COMP | Doc QA |
| Zero Scrolls (Shaham et al., 2023) | 10,392 | - | 4,378 | ✗ | ✓ | ✗ | ✗ | ✗ | ✗ |
| Long Bench (Bai et al., 2023) | 8,120 | - | 4,750 | 350 | ✓ | ✓* | ✗ | ✓* | ✓ |
| L-Eval (An et al., 2023) | 8,008 | 411 | 2,043 | 2,043† | ✓ | ✓ | ✗ | ✗ | ✓ |
| **LooGLE** (Ours) | 19,367 | 776 | 6,448 | 1,101 | ✓ | ✓ | ✓ | ✓ | ✓ |

SUMM refers to Summarization, IR refers to Information Retrieval, TR refers to Timeline Reorder, COMP refers to Computation

* The task is created in a synthetic manner.

† The questions are re-labelled from original data.

We conduct a comprehensive evaluation of 8 representative LLMs on **LooGLE** . We specifically select LLMs that have made significant efforts in addressing the challenge of understanding long contexts as the baselines. The results indicate that better base models with a larger context window size generally achieve better performance. However, all models exhibit poor performance in long dependency tasks, indicating a desperate need to improve the true long dependency understanding capabilities of LLMs. Our dataset serves as an up-to-date benchmark for the cutting-edge assessment and research on the long context understanding and modeling of LLMs.

## 2 RELATED WORK

**Existing models for long context understanding.** There is a growing research interest in developing methods to extend LLMs' context window size, such as utilizing recurrent memory, sparse attention, and external memory. The most popular way is to develop improved transformer architectures (Dong et al., 2023). Efficient transformers (Tay et al., 2020; 2022) are proposed to decrease the memory and time complexity to efficiently model longer texts. Unlike efficient transformers that simplify the attention structure, recurrent transformer (Bulatov et al., 2022) retains the full self-attention mechanism. History information of previous segments is cached and will be leveraged when the subsequent segment is fed into the model without context fragmentation problem. Fine-tuned models on long documents (Wu et al., 2021) are also explored, but they are often effort-costing and face challenges in collecting ground truth fine-tuning data for long text tasks. In addition to approaches developed from modelling and parameter updating perspectives, there are also works incorporating external memory structures and compression techniques for LLMs or utilizing task-oriented process optimization strategies (Gidiotis & Tsoumakas, 2020; Zhou et al., 2022; Ram et al., 2023; Izacard et al., 2022).

**Existing datasets for long context understanding.** There is a growing number of benchmarks proposed to test LLMs' long context understanding ability (Shaham et al., 2023; Li, 2023). Zero-SCROLLS, L-Eval and LongBench are three most recent ones. ZeroSCROLLS (Shaham et al., 2023) automatically processes datasets from different sources into a unified input format with an average of 10k words. However, it mainly focuses on collecting documents and tasks from existing datasets and relies on automatic metrics for limited model comparisons (Shaham et al., 2022). L-Eval (An et al., 2023) differs in re-annotating the data and instructions from similar public datasets with smaller sizes to ensure the quality. Furthermore, it optimizes the evaluation procedures and baselines to obtain more accurate conclusions. LongBench (Bai et al., 2023) provides a bilingual and multi-task dataset featuring diverse sequences of varying lengths, distributions, patterns, languages and domains for a comprehensive evaluation of long context understanding. Nonetheless, it encompasses texts of only thousands of words, out-of-date documents, and tasks mostly restricted to short-term information extraction. Moreover, there are few types of "long dependency" tasks in previous datasets, except for summarization (which LLMs are validated to perform well on) and synthesized tasks like data aggregation and retrieving. To complete those tasks, LLMs solely need to locate pieces of information from the lengthy source input and aggregate them together. In contrast, we propose **LooGLE**  which contains long dependency tasks that are much more challenging, such as event timeline reordering, comprehension/reasoning, and computation. These tasks require not only information retrieval, but also understanding/reasoning over the entire text. We include a detailed comparison with concurrent works in Table 1.

## 3 THE **LooGLE** DATASET

### 3.1 DATASET SELECTION AND CONSTRUCTION

Our **LooGLE**  benchmark consists of three sources: scientific papers, Wikipedia articles, and movie and TV scripts, all covering various topics and categories. These documents are commonly used as corpora in NLP tasks. By replicating the methodology proposed in this paper, they can be collected easily and periodically. All the documents in our **LooGLE**  benchmark are from after 2022 and filtered to be over 10k words in length. We have also considered books, but found that most books meeting our principles are not license-free, therefore, we have excluded them. Statistics of the three sources can be found in Table 2. All the source documents and generated tasks are English-only in this version of the dataset.

Table 2: Statistics of LooGLE

| Dataset | Category | No. Docs | Avg. Words | Max. Words | Min. Words | Avg. Tokens | Task | Subtask | No. Questions |
|---|---|---|---|---|---|---|---|---|---|
| arXiv papers | Physics, Math, Finance, Statistics, Biology, Economics, Computer Science, etc. | 516 | 16,988 | 197,977 | 10,204 | 20,887 | Summarization | — | 516 |
| Wikipedia pages | Events, History, Famous person, Sports, Politics, Arts, Awards, Military, Medical, etc. | 105 | 17,604 | 46,250 | 11,285 | 21,017 | short dependency QA | — | 1,951 |
| | | | | | | | | CR | 152 |
| | | | | | | | | MIR | 158 |
| | | | | | | | long dependency QA | TR | 83 |
| | | | | | | | | COMP | 66 |
| Movie and TV scripts | Action, Adventure, Comedy, Drama, Fantasy, Horror, Mystery, Romantic, Science Fiction, Thriller | 155 | 28,483 | 62,752 | 11,089 | 36,412 | short dependency Cloze | — | 2,880 |
| | | | | | | | | CR | 254 |
| | | | | | | | | MIR | 222 |
| | | | | | | | long dependency QA | TR | 132 |
| | | | | | | | | COMP | 34 |
| Total | | 776 | 19,367 | 197,977 | 10,204 | 24,005 | | | 6,448 |

CR refers to Comprehension & Reasoning, MIR refers to Multiple Information Retrieval, TR refers to Timeline Reorder. COMP refers to Computation.

**arXiv papers**    We pulled data from a massive pool of 10,000 entries on the arXiv website [1] using a random selection method. These entries ranged from January 2022 to April 2023. In the next step, we extracted their abstracts, making them our main source for the summarization task. We were rigorous about maintaining data quality. That meant excluding the reference sections, cleaning up any garbled characters from mathematical equations, and omitting any documents under 10,000 words. After this thorough check, we ended up with a solid collection of 516 research papers.

**Wikipedia articles**    Wikipedia is a free and popular online encyclopedia that provides information and reference on a wide range of topics. Articles are created and edited collaboratively by volunteers from all around the world, making it a dynamic and constantly evolving resource. These Wikipedia articles are perfect for evaluating the long text reading, comprehension, summarization, and information retrieval abilities of LLMs. We firstly downloaded and parsed the most recent page articles present in .bz file format from the official website [2]. Then we kept the articles after 2022 with over 10k words utilizing a subset of the open-source Wikipedia dataset (202203.en) from Hugging Face [3]. Since some pages in the dump file probably no longer exist and redirected to a relevant page, we only retain pages (exempt summaries, citations and references) after redirection.

**Movie and TV scripts**    A movie or TV script typically contains essential information such as scene descriptions, action descriptions, and dialogues between characters. Scripts inherently encapsulate numerous events and facts in dialogue format, necessitating models to deeply comprehend contextual nuances. To comprehend the events unfolding within a dialogue, there is a high demand for reasoning ability, along with the ability to navigate shifts in perspective and grasp the viewpoints of the characters involved. Additionally, scripts are typically lengthy and challenging for LLMs with fixed context window size. All scripts are sourced from three websites [4] [5] [6], consisting of movies and TV shows released after 2022.

### 3.2    MANUAL COMPILATION OF LONG DEPENDENCY QAS

One highlight of our dataset is that we dedicated significant effort to manually compile about 1,100 true long dependency QA pairs. We detail the construction process as follows. Firstly, we randomly sampled a total of 140 long documents from Wikipedia and the scripts dataset. We recruited undergraduate and graduate students from top universities across the nation and organized a manual annotation process to generate long dependency QAs. We categorized long dependency tasks into information retrieval, reading comprehension and reasoning, calculation, and timeline reorder. Each document required a generation of 5 to 10 questions. Additionally, participants were prohibited from employing large language models and tools like ChatGPT for article reading, data generation, and annotation.

For each document, we assigned two separate annotators, one as the questioner and the other as the answerer, who did not know each other's identity to ensure cross-validation quality. A three-step process was conducted to generate questions with accurate and precise answers as well as supporting evidence in the document:

---

[1] https://arxiv.org/

[2] https://dumps.wikimedia.org/

[3] https://huggingface.co/datasets/wikipedia

[4] https://www.scriptslug.com

[5] https://thescriptlab.com/

[6] https://8flix.com

(1) **Question and Answer**: The questioners were tasked with reading the document, formulating questions, providing their own answers, and identifying the evidence in the document to support their answers. The annotation adhered to stringent standards, encompassing the following key principles:

- Long dependency: Each question was required to exhibit a long dependency, i.e., the evidence supporting its answer should have a wide span across the document.
- Diverse problem types: The questioner was required to generate a set of 5 to 10 question-answer pairs for each document. To ensure a balanced question distribution and prevent annotators from generating overly simple questions, this set should not contain more than four questions of the same type.
- Clear and precise questions: The formulation of each question was required to adhere to clarity, conciseness, and no ambiguity, with examples provided.
- Deterministic and objective answers: The answers to the proposed questions were rigorously checked to be deterministic and objective, precluding open-ended ones.

(2) **Answer and Check**: The second step involves the answerers. Each answerer can only access the assigned article text and the posed questions from the questioner in the first step. The answerer was required to thoroughly read the entire document and provide answers to the questions accordingly. The standard for the answers are the same as those for the questioners. In addition to the aforementioned responsibilities, the answerer was also tasked with assessing the quality of the questions, which entails evaluating whether the questions adhere to the standard and whether they are answerable. In instances where a question cannot elicit a definite and unambiguous answer, it is deemed as unsatisfactory, and the answerer is asked to provide constructive feedback for improvement.

(3) **Revise**: In the third step, the questioner had access to the document, the questions, the two sets of answers from both the questioner and the answerer, as well as the feedback from the answerer. The questioner was tasked with first revising the questions based on the feedback, and then consolidating their own answers with those from the answerers to derive the final answers.

In the first step, we acquired a total of 1,137 question-answer pairs. In the second step, 206 of these pairs were identified as non-compliant with the established criteria and were accompanied by suggestions for improvement. The inter-annotator agreement rate is 81.88% (Kim & Park, 2023). Following the revisions conducted in the third step, we ultimately obtained a total of 1,101 high-quality question-answer pairs.

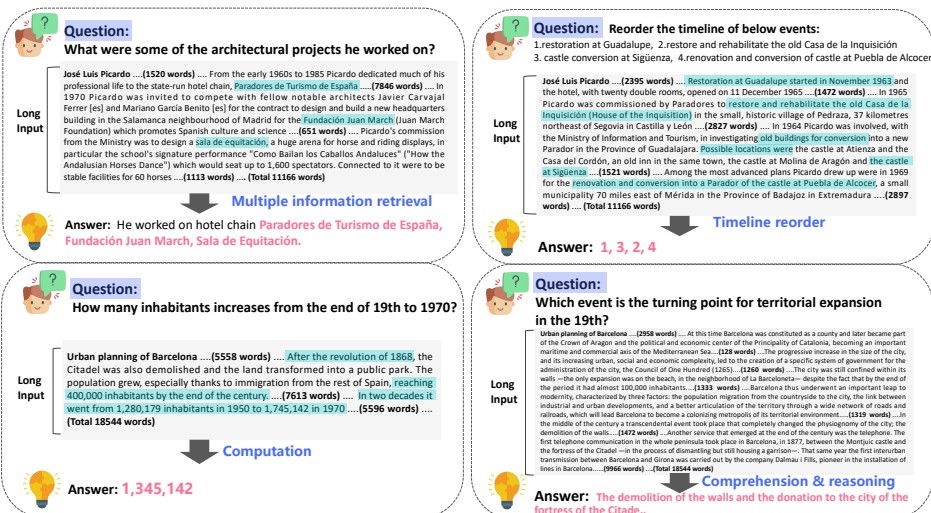

Figure 2: Long dependency QA tasks

## 3.3 TASK DEFINITION

There are two main categories of tasks in **LooGLE** : short dependency and long dependency tasks. For short dependency tasks, we generate short dependency QA from Wikipedia articles and cloze from scripts. For the long dependency tasks, we include summarization for arXiv papers and manually

design QA tasks for long documents understanding. The major subtasks for QA include Multiple information retrieval, Timeline reorder, Computation, Comprehension and reasoning. We delicately generate tasks/questions to customize the intrinsic features of each data source, enhancing the assessment of long context understanding.

### 3.3.1 SUMMARIZATION

For summarization task, we directly utilize the original abstract of each paper as the golden truth without generating it ourselves. The abstracts effectively capture the main content and key information of each paper. The abstract is not employed in any part of the process other than evaluation to ensure fairness in the assessment.

### 3.3.2 LONG DEPENDENCY QA

We manually designed and generated four long dependency tasks: Multiple Information Retrieval, Timeline reorder, Computation, and Comprehension and reasoning. These tasks require advanced capabilities for long context understanding, posing challenges that are valuable for improving LLMs' performance. Examples of long QA can be found in Fig. 2.

- **Multiple Information Retrieval:** Quite different from traditional short term retrieval tasks, this task involves finding multiple and diverse pieces of evidence throughout the entire text for one specific answer. It needs information extraction across widely spanning segments within the long text and aggregate those evidences to obtain the final answer. Evidences are clearly illustrated and can be found directly in the original sentences or sections.
- **Computation:** Similar to the previous task, it first requires multiple information retrieval from a wide range of texts. Most of evidence is numeric, in the format of questions such as "how many times", "how much", "how long", "which number", etc. Further computation is essential to obtain a correct answer based on the deep understanding of the question and exact matching of the related numbers, strongly relying on long context understanding.
- **Timeline reorder:** This task has a more standard format given the instructions "Please reorder the timeline of the below events" by providing several key pieces of information or events with serial numbers. The key information is directly extracted or summarized facts from the source data and the goal is to reorder the given information according to their sequential appearance, spreading widely in the document or sometimes the whole long document.
- **Comprehension and reasoning:** The task requires not only deep comprehension of the question but also complex reasoning to discern what the question truly implies to search for the right evidence. The most frequent patterns for questions can include, but not limited to, queries on reasons for/effect on/contribution to something, attitude/stance of somebody, significance/features, and necessity of some events, and etc. Further comparisons and judgments are required when asking the main/major/highest/most of the evidence mentioned earlier. In addition, answers in this task are not evident from the source. For information retrieval, there are always multi-step reasoning processes involved in modeling the intrinsic correlations and dependencies on the long context before acquiring the answer.

### 3.3.3 SHORT DEPENDENCY TASKS

**Question Answering**    We leverage the powerful language processing and understanding capability of GPT-3.5-turbo to generate short QA pairs from the original text. To ensure compatibility with the maximum input limit of LLM, we truncate each article into several segments while maintaining complete sentences. We then iteratively prompt the LLM with these segments to generate QA pairs, along with their corresponding evidence from the article. The prompt can be found in the Appendix. We manually review the QA pairs and rewrite some of the answers, filtering out non-essential contexts and removing redundant descriptions to guarantee the high quality of QA pairs.

**Cloze**    The goal is to predict the masked entities in the given texts according to the long context. Initially, the script is divided into segments of varying lengths. Then, we employ GPT-3.5-turbo to generate factual summaries align with the source segment using constraints in prompts (see Appendix). Later, we employ BERT-large for Named Entity Recognition (NER) from the generated summaries, limiting the types to person name, location, and organization. Finally, we randomly select

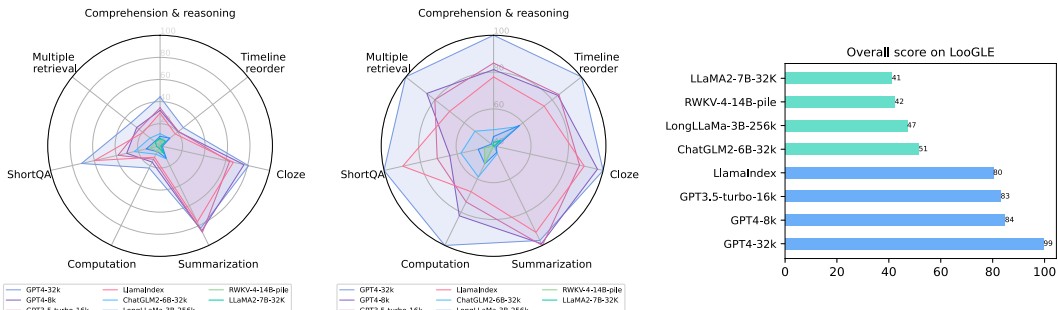

Figure 3: An overview performance of LLMs on **LooGLE** for long context understanding

Table 3: The Performance of Short Dependency QA Task

| Models | Context | Bleu1 | Bleu4 | Rouge1 | Rouge4 | RougeL | Meteor_score | Bert_score | GPT4_score | Exact Match | Partial Match |
|---|---|---|---|---|---|---|---|---|---|---|---|
| | | | | | | Short Dependency QA | | | | Cloze | |
| GPT4-32k | 32k | 24.61 | 11.14 | 61.80 | 50.73 | 60.75 | 32.94 | 78.72 | **71.52** | **70.5** | **80.81** |
| GPT4-8k | 8K | 27.35 | 14.38 | **67.59** | **56.01** | **65.77** | **38.56** | **87.93** | 53.99 | 66.03 | 76.62 |
| GPT3.5-turbo-16k | 16K | 22.67 | 9.62 | 62.56 | 48.63 | 60.66 | 32.58 | 87.04 | 66.82 | 54.64 | 63.42 |
| LlamaIndex | \ | **33.37** | **21.43** | 58.82 | 42.93 | 57.08 | 37.17 | 86.58 | 59.61 | 58.95 | 66.86 |
| ChatGLM-6B-32k | 32k | 14.29 | 6.07 | 20.5 | 13.16 | 20.36 | 13.08 | 87.28 | 23.65 | 0.05 | 0.98 |
| Long_llama-3b | 256k | 1.37 | 0.26 | 26.97 | 11.02 | 26.1 | 11.34 | 71.65 | 13.75 | - | 2.13 |
| RWKV-14b | 8k | 0.8 | 0.04 | 21.7 | 6.39 | 20.64 | 9.41 | 70.42 | 8.93 | - | - |
| Llama2-7b | 32k | 0.18 | 0.00 | 1.86 | - | 1.86 | 1.52 | 61.53 | 3.18 | - | 0.58 |

a certain number (no more than 5) of entities from the summary and mask them as placeholders. Each cloze question can consist of multiple masked entities denoted as "<mask-n>".

## 4 EVALUATION

### 4.1 MODELS SELECTED FOR EVALUATION

**(1) Commercial models:** GPT4-32k, GPT4, and GPT3.5-turbo-16k are all models developed by OpenAI [7]. GPT4-32k can handle up to 32k context input and GPT4 can handel up to 8k context input, GPT3.5-turbo-16k can handle up to 16k context input.

**(2) Open-source models:** LLaMA-2-7B-32K is developed by Together [8] and fine-tuned from Meta's original Llama-2 7B (Touvron et al., 2023) model. It has been expanded to accommodate a context length of 32K using Position Interpolation and it was instruction tuned for summarization and long context QA. ChatGLM2-6B-32K (Du et al., 2022), a product of THUMD, represents an enhancement of the ChatGLM2-6B (Du et al., 2022) model trained with human preference alignment. It is notable for its integration of FlashAttention (Dao et al., 2022) and Positional Interpolation, allowing it to train with an extended context length, increased from 2K to 32K. LongLLaMa (Tworkowski et al., 2023) , derived from openllama, has been fine-tuned using Focused Transformer (Tworkowski et al., 2023) to extend its context to 256k. Lastly, RWKV-4-14B-pile (Peng et al., 2023) is a member of the RWKV model family, notable for its architectural fusion of both Recurrent Neural Networks (RNN) and Transformers. It has been fine-tuned to accommodate a context length of 8K.

**(3) Retrieval Method:** LlamaIndex [9] is a data framework designed for LLMs. It fulfills a dual role by constructing indexes for documents and functioning as a intermediary connecting LLM with data sources. This enables LlamaIndex to retrieve relevant data segments before they are input into the LLM, thereby enhancing the LLM's capacity to effectively handle lengthy text. Here we use the default model text-davinci-003 as the retriever and default chunk size (1024) for LlamaIndex. Based on the conclusions in previous work(Liu et al., 2023a), we artificially truncate the input document to certain sizes (all not larger than the context window size of above mentioned models) by concatenating the head and tail of the input.

---

[7]https://platform.openai.com/docs/models

[8]https://together.ai/

[9]https://github.com/jerryjliu/llama_index

Table 4: The Performance of Long Dependency Tasks

| Models | Context | Bleu1 | Bleu4 | Rouge1 | Rouge4 | RougeL | Meteor_score | Bert_score | GPT4_score |
|---|---|---|---|---|---|---|---|---|---|
| *arXiv Paper Summarization Task* | | | | | | | | | |
| GPT4-32k | 32k | 24.50 | 0.73 | 27.15 | 7.10 | 24.25 | 19.03 | 84.04 | 82.84 |
| GPT4-8k | 8k | **29.02** | **2.09** | **32.08** | **11.11** | 28.85 | **22.64** | 84.92 | 85.42 |
| GPT3.5-turbo-16k | 16k | 28.70 | 1.59 | 32.04 | 10.69 | **28.89** | 22.34 | 84.82 | **86.84** |
| Llama-index | \ | 22.53 | 0.63 | 26.28 | 6.97 | 23.73 | 21.07 | 83.09 | 76.35 |
| ChatGLM-6B-32k | 32k | 0.04 | 0.00 | 5.97 | 0.00 | 5.82 | 6.40 | 73.25 | 13.23 |
| Long_llama-3b | 256k | 4.24 | 0.00 | 4.10 | 0.52 | 3.86 | 3.82 | 73.41 | 12.28 |
| RWKV-14b | 8k | 6.28 | 0.00 | 6.45 | 0.74 | 6.01 | 6.00 | 75.28 | 7.02 |
| Llama2-7b | 32k | 0.03 | 0.00 | 0.12 | - | 0.12 | 0.67 | 71.21 | 7.60 |
| *Long Dependency QA Tasks* | | | | | | | | | |
| GPT4-32k | 32k | 8.55 | 1.40 | **25.59** | 6.36 | **24.04** | **11.13** | 80.16 | **54.09** |
| GPT-4-8k | 8k | **8.94** | 1.01 | 23.45 | 6.57 | 21.69 | 10.18 | 85.36 | 42.12 |
| GPT3.5-turbo-16k | 16k | 6.92 | **1.81** | 25.02 | 6.68 | 23.63 | 10.40 | 83.79 | 45.04 |
| Llama-index | \ | 7.76 | 1.24 | 23.62 | **7.10** | 22.30 | 10.47 | 83.87 | 37.63 |
| ChatGLM2-6B-32k | 32k | 5.62 | 0.01 | 11.94 | 1.45 | 10.84 | 5.55 | **87.18** | 20.64 |
| Long_llama-3b | 256k | 1.04 | 0.00 | 2.96 | 0.03 | 2.71 | 1.66 | 78.60 | 6.48 |
| RWKV-14b | 8k | 0.71 | 0.00 | 18.54 | 1.55 | 17.69 | 3.45 | 71.36 | 5.33 |
| Llama2-7b | 32k | 0.08 | 0.00 | 2.05 | - | 2.05 | 0.46 | 50.28 | 4.18 |

Table 5: The Impact of Context Length

| Models | Context | Bleu1 | Bleu4 | Rouge1 | Rouge4 | RougeL | Meteor_score | Bert_score | GPT4_score |
|---|---|---|---|---|---|---|---|---|---|
| *arXiv Paper Summarization Task* | | | | | | | | | |
| GPT4-32k | 32k | 24.50 | 0.73 | 27.15 | 7.10 | 24.25 | 19.03 | 84.04 | 82.84 |
| GPT4-32k | 24k | 25.57 | 0.81 | 27.61 | 7.53 | 24.73 | 19.86 | 84.07 | 83.15 |
| GPT4-32k | 16k | 24.80 | 0.70 | 27.29 | 7.26 | 24.28 | 19.12 | 84.11 | 82.82 |
| GPT4-32k | 8k | 26.26 | **9.35** | 27.83 | 7.67 | 24.74 | 20.08 | 84.10 | 82.75 |
| GPT4-8k | 8k | **29.02** | 2.09 | **32.08** | **11.11** | **28.85** | **22.64** | **84.92** | **85.42** |
| *Long Dependency QA Tasks* | | | | | | | | | |
| GPT4-32k | 32k | 7.64 | 1.24 | 15.53 | 4.46 | 14.60 | 11.12 | 86.07 | **54.65** |
| GPT4-32k | 24k | 8.23 | 1.66 | 14.92 | 4.12 | 13.90 | 10.60 | 86.16 | 50.61 |
| GPT4-32k | 16k | 8.57 | 1.35 | 16.21 | 4.30 | 14.90 | **11.91** | **86.36** | 47.55 |
| GPT4-32k | 8k | 7.46 | **1.77** | 13.75 | 5.08 | 12.89 | 10.01 | 85.77 | 38.34 |
| GPT4-8k | 8K | **8.94** | 1.01 | **23.45** | **6.57** | **21.69** | 10.18 | 85.36 | 42.12 |

## 4.2 EVALUATION METHODS AND METRICS

**(1) Automatic evaluation** Automatic evaluation metrics can be categorized into two types. Metrics includes Bleu, Rouge, Meteor Score and Bert Score (Li et al., 2023; Mukherjee & Rahman, 2023) are widely used for generative tasks such as summarization and QA. They evaluate tasks mainly based on an n-gram matching and semantic similarity with greater efficiency and cost-effectiveness. For Cloze, Exact Match and Partial Match (Sharma et al., 2023; Engelbach et al., 2023) are employed in our evaluation. Exact Match entails precise comparison between the predicted entities and the ground truth entities while Partial Match allows for fuzzy matching.

**(2) GPT4-as-judgment** Most automatic evaluation metrics are sensitive to semantic expressions, output format, and length. These metrics alone are insufficient for effectively distinguishing between different models. However, recent research has shown that the GPT4 evaluator exhibits high consistency with human evaluation and can serve as a reliable annotator to some extent (Suri et al., 2023; Liu et al., 2023). To provide a more comprehensive assessment of models, we utilize GPT4 as an LLM evaluator to obtain reference results.

**(3) Human evaluation** Human in the loop is necessary for LLM evaluation, especially for free text generation tasks for reference.

## 4.3 RESULTS

Fig. 3 suggests that **LooGLE** provides a more comprehensive evaluation result by integrating various types of short and long dependency tasks. The first radar plot shows the original accuracy evaluated by GPT4 (except cloze) and the partial match (for cloze) among multi-tasks. For better visualization, we scale the score across all models on each task to [40, 100] in the second radar plot and the histogram. Among the 7 major tasks, shortQA, cloze, and summarization can be effectively addressed than other long dependency QA tasks according to the radar plot. GPT4-32k demonstrates its impressive overall performance across all tasks with the highest score. Other commercial model with shorter context length follow behind GPT4-32k with narrow gaps among them. The open-sourced models can hardly

understand the long context and complete the tasks in our benchmark. The empirical results provide insightful conclusions about the multi-task capability of current models in terms of long context comprehension. Detailed evaluation results and further analysis can be seen in the following sections.

### 4.3.1 MAIN RESULTS ON LONG AND SHORT DEPENDENCY TASKS

**Results on short dependency tasks** Table 3 presents the performance (%) of all the baselines on **LooGLE** in short dependency tasks. Notably, GPT4-32k attains the highest accuracy according to the GPT4 evaluator's perspective. GPT4-8k, GPT3.5-turbo-16k, and the retrieval-based LlamaIndex closely follow, demonstrating competitive performance levels. Surprisingly, GPT4-8k exhibits the most robust overall performance in terms of automatic evaluation metrics. It's worth mentioning that GPT4-32k, due to its tendency to generate longer outputs, faces penalties from these automatic metrics. This discrepancy among different metrics highlights the need for improved evaluation methods. Furthermore, in the context of cloze tasks, GPT4-32k excels again when equipped with a longer context window.

**Results on long dependency tasks** Table 4 shows the aggregated results on long dependency tasks. Firstly, we can observe that summarization can be well addressed by commercial models, with GPT-4 evaluation accuracy of over 80%. However, the various types of long dependency QAs in our benchmark apparently pose substantial challenges for current LLMs. Both open-source and commercial models experience a significant performance decline. We will analyze model performance on individual types of QAs in Appendix C. It is validated that longer context window size (thus less information loss due to truncation) indeed helps in long context tasks by comparing GPT4-32k with GPT4-8k. GPT4-8k has a much lower accuracy by answering "The text does not provide information on ..." in many cases. Open-sourced models fall far below the average of commercial models, among which LLaMA2-7B-32K and RWKV-4-14B-pile display almost zero performance. By employing context scaling techniques like positional interpolation and fine-tuning on longer texts, current LLMs can be equipped with much longer context windows than their default limits. Nevertheless, there is still a huge discrepancy between merely increasing the context window size and really understanding the long context. The poor performance on long dependency QAs suggests that we may need to revisit LLMs' long context understanding ability in more challenging tasks other than some simple ones like summarization and retrieval, as they are unable to test whether LLMs understand the inter-dependency in long texts.

### 4.3.2 IMPACT OF CONTEXT LENGTH AND CONTEXT EXTENDING TECHNIQUES

**Impact of varying input length** In Table 5, we study the impact of varying lengths of inputs on long dependency tasks with GPT4 models. We find that expanding input length hardly helps in paper summarization while it substantially enhances the model's performance on long dependency QAs. The difference can be attributed to the inherent nature of the arXiv paper. It has both the introduction and conclusion sections located at the beginning and in the end respectively, which already contain the major sketch of the paper. Meanwhile, in our expectation, longer input promotes the performance of long dependency QAs by introducing less information loss.

## 5 CONCLUSION

This paper introduces a novel benchmark, **LooGLE** , designed to facilitate the assessment of long-context comprehension by LLMs. It addresses the deficiencies in previous datasets by offering considerably longer passages, utilizing relatively new documents after 2022, incorporating multi-source materials from various categories, and notably meticulously designed and annotated tasks with diverse contextual dependencies. Our extensive evaluations unveil substantial limitations in existing LLMs even when provided with considerably extended context windows. The outcomes underscore the utility of our dataset as a valuable reference for evaluating long-context comprehension and present avenues for potential enhancements.

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
