## A  MORE DETAILS OF OUR DATASET

Distributions of the input document length for the whole dataset are shown in Fig. 4. The dependency spanning in words for long dependency QA tasks are shown in 5.

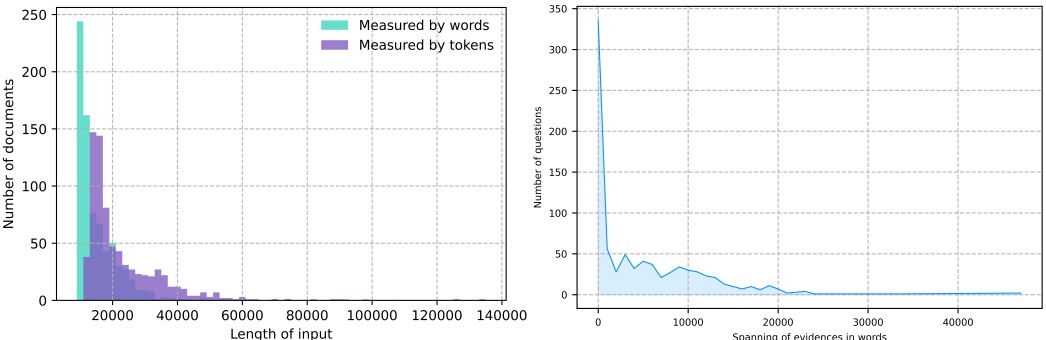

Figure 4: Input document length distributions       Figure 5: Evidence dependency distributions

N-gram sunburst graph for generated QA pairs can be seen in Fig. 6.

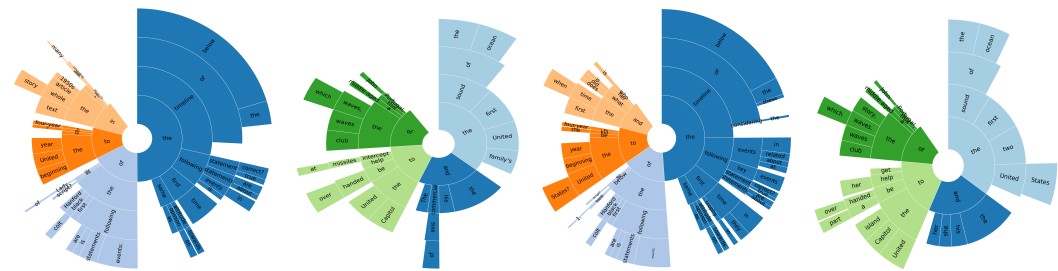

Figure 6: 4-gram sunburst graph for short and long dependency QA. (a) short dependency questions (b) short dependency answers (c) long dependency questions (d) long dependency answers

## B  TASK DEFINITION

The Cloze task formulation process can be seen in Fig. 7.

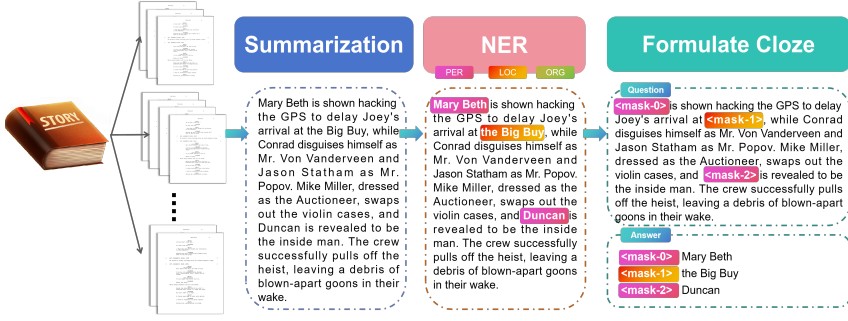

Figure 7: Cloze task

Table 6: Long Dependency QA tasks evaluated by GPT4

| Models | Context | Information Retrieval | Timeline Reorder | Computation | Comprehension and Reasoning |
|---|---|---|---|---|---|
| GPT4-32k | 32k | **33.26** | **26.43** | **22.30** | **44.20** |
| GPT4-8k | 8K | 26.59 | 20.61 | 16.31 | 34.42 |
| GPT3.5-turbo-16k | 16K | 24.05 | 20.88 | 13.49 | 32.10 |
| LlamaIndex |  | 19.38 | 17.23 | 11.43 | 29.53 |
| ChatGLM2-6b-32k | 32k | 11.38 | 10.77 | 8.45 | 10.95 |
| Long_llama-3b | 256k | 6.17 | 11.18 | 5.77 | 8.54 |
| RWKV-14b | 8k | 5.73 | 6.42 | 2.14 | 6.52 |
| Llama2-7b | 32k | 5.49 | 4.73 | 5.71 | 5.28 |

Table 7: The Impact of Context Length on Open-source Model

| Models | Context | Bleu1 | Bleu4 | Rouge1 | Rouge4 | RougeL | Meteor_score | Bert_score | GPT4_score |
|---|---|---|---|---|---|---|---|---|---|
| Long Dependency QA Tasks | | | | | | | | | |
| ChatGLM2-6B-32k | 32k | 5.65 | 0.01 | 11.95 | 1.45 | 10.84 | 5.55 | 87.18 | 20.64 |
| ChatGLM2-6B-32k | 24k | 7.04 | 0.16 | 13.74 | 2.67 | 12.80 | 6.10 | 87.93 | 20.00 |
| ChatGLM2-6B-32k | 16k | 6.37 | 0.01 | 13.26 | 1.49 | 12.29 | 5.69 | 87.87 | **21.53** |
| ChatGLM2-6B-32k | 8k | 4.98 | 0.11 | 11.56 | 1.68 | 10.79 | 5.26 | 87.90 | 21.37 |
| ChatGLM2-6B | 8k | **9.72** | **0.43** | **14.03** | **2.79** | **13.12** | **9.53** | **88.74** | 20.11 |

## C  DEEP DIVE INTO LONG CONTEXT UNDERSTANDING CAPABILITIES

**Individual results on different types of long dependency QAs** Previous research has suggested that simply averaging scores across all tasks may not accurately reflect a model's long context capability, as certain tasks may have a greater influence on the final rank. In this study, we aim to investigate how models perform in specific tasks that require diverse capabilities such as reading comprehension, information retrieval, computation, and reasoning. We leverage GPT 4 as an evaluator using QA accuracy and the results can be seen in Table 6. Among the four tasks, most LLM generally excels in reading comprehension and reasoning tasks and information extraction and comprehension tasks, but lags behind in timeline reorder and computation tasks. We find that there is a great influence on LLMs' performance by the way of questioning. Yes-no questions and multiple choices are easier for LLM to answer with limited searching space rather than open questions in free texts.

**Results on long dependency QAs with/without CoT** To bolster the long-context capabilities of LLMs, we conducted additional experiments designed to unlock their potential using the Chain of Thoughts (CoT) framework (Kojima et al., 2023). We selected LlamaIndex as a representative model, given its impressive performance in both short and long dependency question-answering tasks, alongside strong commercial models such as GPT4. A manual evaluation was carried out on a subset comprising one-third of instances from each task category within long dependency QA. We initiated the LLM with a zero-shot CoT approach, employing prompts such as "Let's think step by step," and furnished a few-shot setup with detailed rationales and standard output formats Wei et al. (2023) to facilitate responses to long dependency questions. As depicted in Fig. 8, the zero-shot CoT approach had minimal impact on accuracy in comprehension and reasoning, as well as multiple retrieval tasks, but yielded a substantial 20% and 10% absolute accuracy increase in timeline reorder and computation. Interestingly, the few-shot CoT approach benefits the first two types but surprisingly leads to a decline in performance in the latter two types compared with zero-shot.We hypothesize the reason is that the evidence and rationales in few-shot examples cannot be generalized to other questions, and including them might on the contrary give wrong guidance to the model.

**Automatic evaluation on timeline reorder** In order to evaluate the performance of time reorder task outputs, it is essential to address discrepancies arising from the diverse formats produced by

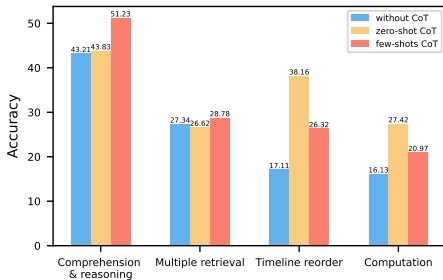

Figure 8: Long dependency QAs w/o CoT        Figure 9: Output distributions on QA tasks

Table 8: The Performance of Timeline Reorder Task

| Models | LSD | LMD | SD | SDD | LSD-S | LMD-S | SD-S | SDD-S | Non-standard(%) |
|---|---|---|---|---|---|---|---|---|---|
| GPT4-32k | **1.04** | **0.57** | **0.93** | **1.1** | 1.21 | **0.82** | 1.41 | 1.60 | 52.80 |
| GPT4-8k | 1.24 | 0.64 | 1.04 | 1.281 | 1.43 | 0.92 | 1.51 | 1.74 | 49.31 |
| LlamaIndex | 1.55 | 0.78 | 1.19 | 1.551 | 1.95 | 1.08 | 1.65 | 2.09 | **39.72** |
| GPT3.5-turbo-16k | 3.58 | 1.43 | 2.17 | 2.916 | **1.05** | 0.86 | 1.24 | **1.26** | 77.21 |
| Long_llama-3b | 4.18 | 1.71 | 2.59 | 3.30 | 1.90 | 1.12 | 1.80 | 2.07 | 92.92 |
| ChatGLM2-6B-32k | 4.31 | 1.74 | 2.63 | 3.37 | 1.83 | 1.17 | 1.50 | 2.00 | 99.07 |
| RWKV-14b | 4.33 | 1.75 | 2.64 | 3.38 | 1.90 | 0.97 | **1.00** | 1.75 | 98.13 |
| Llama2-7b | 4.33 | 1.75 | 2.64 | 3.38 | 2.50 | 1.17 | 1.33 | 2.33 | 98.60 |

various models. Typically, these outputs comprise conventional numerical sequences, but errors in non-standard formats when evaluation necessitate preprocessing for accurate assessment. A proposed approach involves converting the serial numbers in the candidate answers from their original question into Roman numbers (*i.e.*, I, II, · · · ), thereby enhancing discrimination through regular expression matching. Four key metrics, namely, LSD (location square deviation), LMD (location mean deviation), SD (swap deviation), and SDD (swap distance deviation), are employed to measure the similarity of numeric sequences, refer to Appendix D for metric details. Smaller deviations indicate a higher degree of resemblance between the sequences. Any outputs that are empty, possess unequal lengths, or contain extra elements are categorized as non-standard. The maximum deviation between the provided ground truth and all corresponding candidate answers is computed as the worst score for evaluation purposes. The percentage of non-standard outputs for each model and corresponding performances can be found in Table 8. As seen, it is evident that except for GPT4, which demonstrates a remarkable degree of adherence and alignment following Reinforcement Learning from Human Feedback (RLHF) (Lee et al., 2023) , most open-sourced models struggle to generate texts in the correct format with less than 10%. However, this issue can be mitigated in significantly large models through the utilization of few-shot prompts and mandatory instructions. This phenomenon results in performance penalties when assessed using automated metrics. Consequently, to ascertain the genuine capacity of LLMs in this task, we calculate the four metrics exclusively for outputs in standard format ("-S").

**The Impact of Context Length on open-source model** In order to exlpore the impact of Context Length on open-source model, we further compare the varying input lengths of ChatGLM2-6B-32k (32k context window) with ChatGLM2-6B (8k context window) on long dependency QA. We select ChatGLM2 as the testing model since it achieves better performance in our previous experiments among different open-sourced models. The detailed reault can be found in Table 7. We find that the original version model ChatGLM2-6B performs better than ChatGLM2-6B-32k across the automated metrics given the same length of inputs. There is a performance decline for the model when extending the longer context window. It can be attributed to the information loss introduced by the scaling techniques of long context models, which calls for further improvement on open-sourced LLMs. Moreover, for ChatGLM2-6B-32k with varying lengths of inputs, we find that the extension of longer inputs has limited impact on performance. It is because the long dependency QAs in our dataset request for the long dependency comprehension and modeling capability, along with further computation and reasoning. Our dataset propose higher demands on true long context undetstanding, which needs to be desperately resolved and enhanced for further LLMs.

**Discrepancy in generated outputs of models** Distributions of generated outputs of various models are depicted in Fig. 9. It is noteworthy that well-behaved models consistently produce shorter responses, averaging around 50 words, irrespective of the question type, particularly in short-term question answering scenarios. In contrast, models fine-tuned with longer textual inputs, such as LLaMA2-7B-32K, RWKV-4-14B-pile, and LongLLaMa-3B, tend to yield significantly lengthier responses, even when a maximum generation constraint of 500 tokens is enforced. An interesting deviation is observed in LongLLaMa-3B, which demonstrates variability in response lengths across both tasks. This behavior may stem from challenges in comprehending and addressing exceedingly complex long question-answering tasks. Consequently, the model appears to prioritize extracting a maximum number of pertinent contexts from its memory to generate sufficiently extensive responses that are deemed acceptable and rational.

Moreover, upon closer manual examination of model outputs, a significant disparity in generation quality is observed across various LLMs and task types, indicating a non-specific issue. Notably,

Table 9: Bad cases study on the long dependency QA

| Long QA Tasks | Hallucination[*] | Redundant retrieval[†] | Insufficient retrieval[*] | Irrelevant answer[◇] | No relevant context[∧] | Wrong/No reasoning[×] | Others |
|---|---|---|---|---|---|---|---|
| Computation | 31.11 | 24.44 | 15.56 | 0.00 | 20.00 | 0.00 | 8.88 |
| Multiple information retrieval | 14.71 | 31.37 | 28.43 | 13.73 | 13.73 | 0.00 | 7.84 |
| Comprehension and reasoning | 14.29 | 10.99 | 21.98 | 18.68 | 16.48 | 10.99 | 6.59 |

[*] Evidence of predictions is not shown up in the original inputs and generated by LLM itself from nowhere.
[†] Apart from the right evidence, irrelevant evidence is also redundantly retrieved.
[*] Not all of the essential evidence to answer the question is retrieved.
[◇] Evidence of predictions generated have no or minor correlation with the question.
[∧] No relevant context in LLM's memory and refuse to answer the question.
[×] Fail to retrieve evidence that needs further reasoning other than directly extracting from the inputs.

commercial models like GPT4, GPT3.5, and LlamaIndex consistently generate outputs that exhibit a higher degree of human-likeness, completeness, and logical coherence within a structured format. These models consistently deliver contextually relevant, query-based responses. In contrast, open-sourced models, such as ChatGLM2-6B-32k, tend to offer shorter answers, occasionally confined to numeric responses. In cases where a definite answer is lacking, ChatGLM2-6B-32k compensates by retrieving relevant contextual information. However, the RNN-based model RWKV-4-14B-pile often generates duplicated responses or resorts to repeating the given questions to reach the maximum token length, sometimes resorting to code generation to address issues related to its training data. The performance of the LLaMA2-7B-32K model is notably worse, as it sporadically produces irrelevant or nonsensical text, along with the inclusion of special symbols when it fails to provide meaningful answers. More examples of outputs from different models can be seen in Appendix G.

**Probable explanations for long QA bad cases** To investigate whether the models have effectively memorized and comprehended lengthy contextual information, we conducted a comprehensive manual analysis of the underlying causes of failures in each long question-answering task. The rationale behind CoT analysis aided in understanding how models decompose and tackle challenges associated with extended dependency-based QA. From Table 9, Our observations reveal that LLMs struggle with these tasks primarily due to their inability to extract precise information and a propensity to generate responses that lack factual accuracy. Constraints imposed by the inherent context window limitations, coupled with information loss resulting from the optimized Transformer and position encoding, contribute to their struggles in memorizing the original extensive contexts. In most cases, models attempt to compensate by retrieving and integrating the most pertinent evidence, even if it results in redundant answers. However, they also acknowledge their insufficient context and, at times, abstain from providing responses rather than resorting to nonsensical answers. Furthermore, addressing these challenges necessitates enhanced comprehension and reasoning abilities, particularly when answers are not clearly evident across multiple pieces of evidence scattered throughout the raw texts. The insights derived from our benchmark analysis offer a scientific foundation and pave the way for promising research directions aimed at augmenting LLM capabilities for handling long contextual inputs. These findings underscore the need for further progress in comprehension, computation, and reasoning tasks using our dataset to effectively enhance LLMs' capacity to understand extended dependency contexts.

## D  TIMELINE REORDER EVALUATION METRICS

We employ 4 metrics to measure the similarity of numeric output sequences for timeline reorder tasks. For given two numeric sequences $A$ and $B$ with the same sequence length $n$, $i[A]$ and $i[B]$ is the $i$th number in each sequence. They can be computed using the formula below:
LSD is the abbreviation for location square deviation:

$$LSD(A, B) = \frac{1}{n} \sum_{i=0}^{n-1} (i[A] - i[B])^2 \qquad (1)$$

LMD is the abbreviation for location mean deviation:

$$LMD(A, B) = \frac{1}{n} \sum_{i=0}^{n-1} |i[A] - i[B]| \tag{2}$$

SD is the abbreviation for swap deviation:

$$SD(A, B) = min(W(A \to B) = min(\sum_{s \in A \to B} 1) \tag{3}$$

SDD is the swap distance deviation:

$$SDD(A, B) = min(W(A \to B) = min(\sum_{s \in A \to B} |i - j|) \tag{4}$$

where $s = A(i, j)$ means the swap between the $i$th and $j$th element in $A$. $S = A \to B$ means a series of swap actions to convert $A$ to $B$. $W(S) = \sum_{s \in S} w(s)$ means the weights sum of all the swap actions in $S$, where $w(s) = 1$ in SD and $w(s) = |i - j|$ in SSD.

## E  PROMPTS

### E.1  SHORT DEPENDENCY QA PAIR GENERATION

> [seg] = {*Input long texts*}
> Please generate 2 questions and corresponding answers based on given [seg] in less words as possible. Return reference text S, question Q and answer A from [seg] in json format as: {"S": ,"Q": ,"A": },{"S": ,"Q": ,"A": }.

### E.2  SHORT AND LONG DEPENDENCY QUESTION AND ANSWERING

> **Instruction:** Please answer the question based on the given long texts below.
> {*Input long texts*}
> Question: {*Question*}
> Answer:

### E.3  SCRIPTS SEGMENT SUMMARIZATION FOR CLOZE FORMULATION

> **Instruction:** Please write a summary for this script segment within 500 words, focusing on describing objective facts and avoiding subjective opinions.
> {*scripts segement*}
> Summary:

### E.4  CLOZE

> example= {"<mask-0>": "Bob", "<mask-1>": "Gorrosion Magazine", "<mask-2>": "Bethel Horizon"}
> **Instruction:** Please answer the cloze question based on the given long texts below. Each of the placeholder "<mask-n>" in the question could be an entity of Person, Location or Organization. The same masks represent the same entity. Output a json format answer, for example:{*example*}
> {*Input long texts*}
> Question: {*Question*} *What are the masked entities?*
> Answer:

### E.5 SUMMARIZATION

> **Instruction:** Please generate a summary of the below paper.
> *{Input long texts}*
> Summarization:

### E.6 TIMELINE REORDER

> **Instruction:** Please answer the question based on the given long texts below. Please use Roman numbers only for answers.
> *{Input long texts}*
> Question: *{Question}*
> Answer:

### E.7 QA TASK EVALUATION BY LLM (GPT4)

> **Instruction:** Given one question, there is a groundtruth and a predict answer. Please decide whether they are the same or not in semantic. Please only output True or False.
> Question: *{Question}*
> groundtruth = *{Reference answer}*
> predicted answer = *{Generated output}*

### E.8 SUMMARIZATION TASK EVALUATION BY LLM (GPT4)

> **Instruction:** Please compare predicted summary with the goundtruth and evaluate the predicted summary from the perspectives of information completeness, consistency, fluency, and grammar by giving a score within the range of 0 to 100.
> groundtruth = *{Reference answer}*
> predicted summary = *{Generated output}*

### E.9 FEW-SHOT COT FOR LONG QA

> **Instruction:** Please answer the question based on the given long texts below.
> *{Input long texts}*
> *{Demonstrations}*
> Question: *{Question}*
> Answer:

### E.10 ZERO-SHOT COT FOR LONG QA

> **Instruction:** Please answer the question based on the given long texts below.
> *{Input long texts}*
> Question: *{Question}*
> Answer: Let's think step by step.

## F    EXAMPLES FOR LONG CONTEXT UNDERSTANDING TASKS

### F.1    SHORT DEPENDENCY QUESTION AND ANSWERING

---

**Question:**
Who did Picardo collaborate with for building preservation and restoration projects?
**Evidence:**
On qualifying in 1951, Picardo pursued his interest in historical architecture by collaborating on a number of building preservation and restoration projects with the Spanish architect and architectural historian Fernando Chueca Goitia, who was 8 years his senior.
**Answer:**
He collaborated with Spanish architect and architectural historian Fernando Chueca Goitia.

---

**Question:**
What was the nickname given to the 18th century period?
**Evidence:**
The 18th century was nicknamed the 'Age of Enlightenment', as it was the period in which the Enlightenment emerged, a philosophical movement that defended reason and science against religious dogmatism.
**Answer:**
The Age of Enlightenment.

---

### F.2    CLOZE

---

**Question:**
<mask-0>  is shown hacking the GPS to delay Joey's arrival at <mask-1>, while Conrad disguises himself as Mr. Von Vanderveen and Jason Statham as Mr. Popov. Mike Miller, dressed as the Auctioneer, swaps out the violin cases, and <mask-2>is revealed to be the inside man. The crew successfully pulls off the heist, leaving a debris of blown-apart goons in their wake.
**Long input:**
When a caper crew needs something blown up for a heist, they call upon The Demolition Expert. They are often minor characters who are not given much screen ....(104,094 words).... Joey is driving to the Big Buy, always craning back... like there's a phantom on his tail. Suddenly, the GPS chimes. GPS VOICE Rerouting . DRIVER Shit. Uh, boss, it says it just added twenty minutes. The speed past – A BLACK MUSTANG parked in a turnaround. Mary Beth in the driver's seat, clacking away on a laptop, hacking the GPS . ....(150 words)....we couldn't have done it without Duncan– Reveal Duncan , smiling big. He raises his glass. FLASH: DUNCAN and TWO MORE GOONS hurry around the corner of the STADIUM HALLWAY and stop dead in their tracks when they see – A debris of blown-apart Goons littering the hallway. ....(2,670 words).
**Answer:**
{"<mask-0>": "Mary Beth",  "<mask-1>": "the Big Buy",  "<mask-2>": "Duncan"}

---

### F.3    SUMMARIZATION

---

**Long input:**
Distinction and quadratic base change for regular supercuspidal representations Chuijia Wang 1 Introduction Let be a connected reductive algebraic group over a non-archimedean local field with residual characteristic ....(21,000 words).... Basically, one can describe all the characters of which occur in in terms of certain intersection property between the Kostant sections of and the orbit of the generic element associated to. ....(500 words).
**Summary:**
In this article, we study Prasad's conjecture for regular supercuspidalrepresentations based on the machinery developed by Hakim and Murnaghan tostudy distinguished representations, and

---

the fundamental work of Kaletha onparameterization of regular supercuspidal representations. For regularsupercuspidal representations, we give some new interpretations of thenumerical quantities appearing in Prasad's formula, and reduce the proof to thecase of tori. The proof of Prasad's conjecture then reduces to a comparison ofvarious quadratic characters appearing naturally in the above process. We alsohave some new observations on these characters and study the relation betweenthem in detail. For some particular examples, we show the coincidence of thesecharacters, which gives a new purely local proof of Prasad's conjecture forregular supercuspidal representations of these groups. We also prove Prasad'sconjecture for regular supercuspidal representations of G(E), when E/F isunramified and G is a general quasi-split reductive group.

## F.4 MULTIPLE INFORMATION RETRIEVAL

**Question:**
What were some of the architectural projects José Luis Picardo worked on?
**Long input:**
José Luis Picardo ....(1,520 words) .... From the early 1960s to 1985 Picardo dedicated much of his professional life to the state-run hotel chain, Paradores de Turismo de España .....(7,846 words) .... In 1970 Picardo was invited to compete with fellow notable architects Javier Carvajal Ferrer [es] and Mariano García Benito [es] for the contract to design and build a new headquarters building in the Salamanca neighbourhood of Madrid for the Fundación Juan March (Juan March Foundation) which promotes Spanish culture and science ....(651 words) .... Picardo's commission from the Ministry was to design a sala de equitación, a huge arena for horse and riding displays, in particular the school's signature performance "Como Bailan los Caballos Andaluces" ("How the Andalusian Horses Dance") which would seat up to 1,600 spectators. Connected to it were to be stable facilities for 60 horses ....(1,113 words).
**Answer:**
He worked on hotel chain Paradores de Turismo de España, Fundación Juan March, Sala de Equitación.
**Explanation:**
Based on the deep understanding of given question, we need to extract all the evidence of architectural projects José Luis Picardo have worked on. There are total three works spreading in the original inputs independently as shown above.

## F.5 TIMELINE REORDER

**Question:**
Reorder the timeline of below events:
1.restoration at Guadalupe,
2.restore and rehabilitate the old Casa de la Inquisición
3.castle conversion at Sigüenza,
4.renovation and conversion of castle at Puebla de Alcocer
**Long input:**
José Luis Picardo ....(2,395 words) .... Restoration at Guadalupe started in November 1963 and the hotel, with twenty double rooms, opened on 11 December 1965 ....(1,472 words) .... In 1965 Picardo was commissioned by Paradores to restore and rehabilitate the old Casa de la Inquisición (House of the Inquisition) in the small, historic village of Pedraza, 37 kilometres northeast of Segovia in Castilla y León ....(2,827 words) .... In 1964 Picardo was involved, with the Ministry of Information and Tourism, in investigating old buildings for conversion into a new Parador in the Province of Guadalajara. Possible locations were the castle at Atienza and the Casa del Cordón, an old inn in the same town, the castle at Molina de Aragón and the castle at Sigüenza ....(1,521 words) .... Among the most advanced plans Picardo drew up were in 1969 for the renovation and conversion into a Parador of the castle at Puebla de Alcocer, a small municipality 70 miles east of Mérida in the Province of Badajoz in Extremadura ....(2,897 words).
**Answer:**



1, 3, 2, 4

**Explanation:**

The four events provided in the question sequentially happen with thousands of words spanning. We firstly locate the exact sentences describing the event in the original inputs above. Then we reorder them based on the their occurrence.

</div>

### F.6 COMPUTATION



**Question:**

How many inhabitants increases from the end of 19th to 1970?

**Long input:**

Urban planning of Barcelona ....(5,558 words) .... After the revolution of 1868, the Citadel was also demolished and the land transformed into a public park. The population grew, especially thanks to immigration from the rest of Spain, reaching 400,000 inhabitants by the end of the century. ....(7,613 words) .... In two decades it went from 1,280,179 inhabitants in 1950 to 1,745,142 in 1970 ....(5,596 words).

**Answer:**

1,345,142

**Explanation:**

Firstly, we locate the numeric of inhabitants which only appear between 1900 to 1970 from the input as evidence. There are three relevant numbers: 400,000, 280,179 and 1,745,142. Then we make computation on 1,745,142 - 400,000 = 1,345,142 to get the final answer.

</div>

### F.7 COMPREHENSION AND REASONING



**Question:**

Which event is the turning point for territorial expansion in the 19th?

**Long input:**

Urban planning of Barcelona ....(2,958 words) .... At this time Barcelona was constituted as a county and later became part of the Crown of Aragon and the political and economic center of the Principality of Catalonia, becoming an important maritime and commercial axis of the Mediterranean Sea....(128 words) ....The progressive increase in the size of the city, and its increasing urban, social and economic complexity, led to the creation of a specific system of government for the administration of the city, the Council of One Hundred (1,265)....(1,260 words) ....The city was still confined within its walls —the only expansion was on the beach, in the neighborhood of La Barceloneta— despite the fact that by the end of the period it had almost 100,000 inhabitants....(1,333 words) ....Barcelona thus underwent an important leap to modernity, characterized by three factors: the population migration from the countryside to the city, the link between industrial and urban developments, and a better articulation of the territory through a wide network of roads and railroads, which will lead Barcelona to become a colonizing metropolis of its territorial environment.....(1,319 words) ....In the middle of the century a transcendental event took place that completely changed the physiognomy of the city; the demolition of the walls.....(1,472 words) ....Another service that emerged at the end of the century was the telephone. The first telephone communication in the whole peninsula took place in Barcelona, in 1877, between the Montjuic castle and the fortress of the Citadel —in the process of dismantling but still housing a garrison—. That same year the first interurban transmission between Barcelona and Girona was carried out by the company Dalmau i Fills, pioneer in the installation of lines in Barcelona......(9,966 words).

**Answer:**

The demolition of the walls and the donation to the city of the fortress of the Citade.

**Explanation:**

There is no exact answer that can be extracted directly from the original input and further reasoning is essentially needed. Firstly, we find all the relevant sentences or paragraphs on the territorial expansion which are in the 19th. Temporal events and key words act as clues for sorting out the evidence. Among the evidence, some are factual plots while others are critical

</div>

comments. Based on further comprehension, we need to compare the various factors of territorial expansion underlying the evidence and decide the most decisive and influential turning point events to answer the question.

## G    EXAMPLES OF GENERATED OUTPUTS FOR MODELS

### G.1    GPT4-32K

1. short QA

**Question:** What were the charges against the two members of the Rock Machine Perth chapter?
**Output:** The two members of the Rock Machine Perth chapter, Kyle Barry and Brent Reker, were charged with extortion and demanding property with oral threats.
**Groundtruth:** Extortion and demanding property with oral threats.

**Question:** Why did the federal government create relocation programs for Native Americans?
**Output:** The federal government created relocation programs for Native Americans in an effort to expand their employment opportunities. Many Native Americans were living in rural and isolated areas, and the government believed that moving them to urban areas would offer better job prospects. This was part of a wider policy shift towards giving tribes more authority for self-determination.
**Groundtruth:** To expand their employment opportunities.

2. long QA

**Question:** According to World Bank's estimation, what is the difference between the amount of World Bank's aid to Turkey and the amount of damaged suffered by Syria?
**Output:** The World Bank estimated $5.1 billion in damages suffered by Syria and it provided $1.78 billion in aid to Turkey. So, the difference is $3.32 billion.
**Groundtruth:** 3.32 Billion.

**Question:** What efforts did guy make to improve his life?
**Output:** Guy made several efforts to improve his life, including attending therapy sessions, building a tree house for his nephews, discussing his feelings and insecurities with his friends, and standing up for himself against his aggressive father. He also expressed his feelings to his ex-girlfriend Jen, trying to reconcile with her and offering her a house he planned to renovate. He made attempts to change his negative self-talk and confront his past traumas.
**Groundtruth:** 1. Receive psychotherapy: Guy chooses to see a psychiatrist to face his own problems, which is an important step for him to change. 2. Introspection: Guy gradually enlightened himself during the treatment, facing the relationship problems between him, his father and his friends. 3. Write a letter to Jen: Guy wrote a sincere letter to Jen, expressing his change of attitude, trying to win back Jen's heart. 4. Reaching out to old friends: Despite the embarrassment, Guy reached out to old friends and tried to work things out. 5. Renovate the house: Guy takes on the job of renovating the house to plan for his future life. 6. Dealing with the relationship with the father: Guy faces his father, expresses his thoughts, and no longer avoids this issue. 7. Change yourself: Guy works hard, writes letters, receives treatment, etc., showing his determination to change himself.

3. summarization

**Output:** This paper discusses the distinction and quadratic base change for regular supercuspidal representations. The author focuses on understanding Prasad's conjecture on the Galois distinction problem, which proposes a relationship between the distinction property of an irreducible

representation and the base change property of its Langlands parameters. The author's work highlights that even though significant progress has been made in proving the conjecture, a general method that works for all Galois symmetric pairs would be interesting. It extends the understanding of the relationship between distinguished representations and Langlands functoriality. The results support the idea of using a purely local approach to prove the conjecture for regular supercuspidal representations without constraints on the Galois symmetric pair. The author also outlines strategies to improve the exact analysis of terms appearing in Prasad's identity. The article provides evidence supporting the conjecture by evaluating particular examples.

**Groundtruth:** In this article, we study Prasad's conjecture for regular supercuspidal representations based on the machinery developed by Hakim and Murnaghan to study distinguished representations, and the fundamental work of Kaletha on parameterization of regular supercuspidal representations. For regular supercuspidal representations, we give some new interpretations of the numerical quantities appearing in Prasad's formula, and reduce the proof to the case of tori. The proof of Prasad's conjecture then reduces to a comparison of various quadratic characters appearing naturally in the above process. We also have some new observations on these characters and study the relation between them in detail. For some particular examples, we show the coincidence of these characters, which gives a new purely local proof of Prasad's conjecture for regular supercuspidal representations of these groups. We also prove Prasad's conjecture for regular supercuspidal representations of G(E), when E/F is unramified and G is a general quasi-split reductive group.

4. cloze

**Question:** The script segment of " Wildfire 2022" takes place in Tulare County, California, where the sky is grey and ashy due to the nearby fires. The story follows <mask-1>, a seven-year-old mute girl, who runs to school along a busy road. <mask-0>, a Trans Latin Woman, is seen singing and dancing to " Opposites Attract" while trying to fix her old car. Lu goes to school and stands in line for lunch, where she is ignored by other kids. She pays for her lunch in change and sits alone at the back of the cafeteria. Later, Fireman <mask-2>speaks to Mrs. Consuelos' third-grade classroom about fire safety, and Lu is chosen to answer a question despite not speaking. The leading cause of wildfires is discussed, and it is revealed that humans are the leading cause of wildfires due to accidents like leaving an open campfire burning or tossing a cigarette out a car window.
**Output:** {"<mask-0>": "Merribelle", "<mask-1>": "Lu", "<mask-2>": "Josh"}
**Groundtruth:** {"<mask-0>": "Merribelle", "<mask-1>": "Lu", "<mask-2>": "Josh"}

### G.2   GPT4-8K

1. short QA

**Question:** When did large cattle ranches start to appear in the area?
**Output:** As early as 1734.
**Groundtruth:** As early as 1734.

**Question:** What contributed to the economic growth of Rio Grande do Sul at the end of the 19th century?
**Output:** The economic growth of Rio Grande do Sul at the end of the 19th century was contributed by the growth of trade, the arrival of immigrants of other origins such as Italians and Jews, and the establishment of a growing industry and a rich bourgeois class. The state had become the third largest economy in Brazil by the turn of the 20th century.
**Groundtruth:** Trade and immigration of Italians and Jews.

2. long QA

**Question:** Reorder the timeline of below events:
1. Guardian Reported Sexually Assault.
2. Nunez Announced taking place An Internal Investigation.
3. A Young Woman Was Reported To Have Lost Her Thumb.
**Output:**
3. A Young Woman Was Reported To Have Lost Her Thumb.
1. Guardian Reported Sexually Assault.
2. Nunez Announced taking place An Internal Investigation.
**Groundtruth:** 2,1,3

**Question:** Was Maurice Duplessis' treatment of the news publishing industry consistent or inconsistent over his two terms?
**Output:** The text does not provide specific information on Maurice Duplessis' treatment of the news publishing industry over his two terms.
**Groundtruth:** Inconsistent.

3. summarization

**Output:** This paper by Chuijia Wang focuses on the study of representation theory of a connected reductive algebraic group over a non-archimedean local field. The main focus is to understand Prasad2019s conjecture on the Galois distinction problem. The author provides evidence on the possibility of proving this conjecture for regular supercuspidal representations without constraints on the Galois symmetric pair using a different purely local method. The paper also discusses the work of Sakellaridis and Venkatesh, Dipendra Prasad, and others in the field. The author proves Prasad2019s conjecture for regular supercuspidal representations of certain groups and provides a detailed study of these quadratic characters. The paper concludes with a discussion on the distinction problem and the construction of local Langlands correspondence.
**Groundtruth:** In this article, we study Prasad's conjecture for regular supercuspidal representations based on the machinery developed by Hakim and Murnaghan to study distinguished representations, and the fundamental work of Kaletha on parameterization of regular supercuspidal representations. For regular supercuspidal representations, we give some new interpretations of the numerical quantities appearing in Prasad's formula, and reduce the proof to the case of tori. The proof of Prasad's conjecture then reduces to a comparison of various quadratic characters appearing naturally in the above process. We also have some new observations on these characters and study the relation between them in detail. For some particular examples, we show the coincidence of these characters, which gives a new purely local proof of Prasad's conjecture for regular supercuspidal representations of these groups. We also prove Prasad's conjecture for regular supercuspidal representations of G(E), when E/F is unramified and G is a general quasi-split reductive group.

4. cloze

**Question:** The script segment of " Wildfire 2022" takes place in Tulare County, California, where the sky is grey and ashy due to the nearby fires. The story follows <mask-1>, a seven-year-old mute girl, who runs to school along a busy road. <mask-0>, a Trans Latin Woman, is seen singing and dancing to " Opposites Attract" while trying to fix her old car. Lu goes to school and stands in line for lunch, where she is ignored by other kids. She pays for her lunch in change and sits alone at the back of the cafeteria. Later, Fireman <mask-2>speaks to Mrs. Consuelos' third-grade classroom about fire safety, and Lu is chosen to answer a question despite not speaking. The leading cause of wildfires is discussed, and it is revealed that humans are the leading cause of wildfires due to accidents like leaving an open campfire burning or tossing a cigarette out a car window.
**Output:** {"<mask-0>": "Merribelle", "<mask-1>": "Lu", "<mask-2>": "Josh"}
**Groundtruth:** {"<mask-0>": "Merribelle", "<mask-1>": "Lu", "<mask-2>": "Josh"}

## G.3 GPT3.5-turbo-16k

1. short QA

> **Question:** What was unveiled by Buckingham Palace for the Coronation of Charles III and Camilla?
> **Output:** The official Coronation logo was unveiled by Buckingham Palace for the Coronation of Charles III and Camilla.
> **Groundtruth:** The official Coronation logo.

> **Question:** What was the percentage decline in house prices in the UK?
> **Output:** There is no specific information provided in the text about the percentage decline in house prices in the UK.
> **Groundtruth:** 3.4%.

2. long QA

> **Question:** Which church hasn't been built during the Phase 1 of church construction?
> 1. The Main Church of Sabará
> 2. The Cathedral Basilica of Salvador
> 3. The Church of Our Lady of Grace
> **Output:** The Main Church of Sabará.
> **Groundtruth:** 1,2.

> **Question:** Do Reader's Digest and Alexander Cockburn agree on Mujahideen?
> **Output:** No, Reader's Digest and Alexander Cockburn do not agree on the Mujahideen. Reader's Digest generally portrayed the Mujahideen as freedom fighters and heroes, while Cockburn criticized the US support for the Mujahideen and argued that they were not the noble fighters they were made out to be.
> **Groundtruth:** No.

3. summarization

> **Output:** This paper focuses on understanding Prasad's conjecture on the Galois distinction problem for regular supercuspidal representations. The distinction problem refers to the question of determining which irreducible representations of a reductive group are distinguished with respect to a pair of subgroups and a character. The paper provides evidence for the possibility of proving this conjecture for regular supercuspidal representations using a different purely local method.
> The paper begins by introducing the Langlands-Vogan bijection, which relates irreducible representations of a reductive group to Langlands parameters. It then discusses the local Langlands correspondence for regular supercuspidal representations and the parametrization of these representations using regular supercuspidal packet data. The paper also reviews the character expansions and the Murnaghan-Kirillov formula for regular semisimple and nilpotent orbits.
> The main part of the paper focuses on the computation of various characters associated with regular supercuspidal representations. It considers the characters associated with Prasad's conjecture, Hakim's character, and the character associated with - data. The paper provides explicit computations for these characters in various cases, such as when the quadratic extension is unramified or ramified, and when the group is odd or even. The paper also compares these characters and shows that they satisfy the necessary conditions for Prasad's conjecture.
> The paper concludes by providing examples of regular supercuspidal representations of specific groups, such as and , and showing that Prasad's conjecture holds for these representations. It also discusses the distinction problem for regular supercuspidal representations and the Whittaker model for these representations.
> Overall, this paper provides a detailed analysis of Prasad's conjecture for regular supercuspidal

representations and presents new evidence for its validity. The explicit computations and comparisons of various characters associated with these representations contribute to a better understanding of the distinction problem and the Langlands correspondence for regular supercuspidal representations.

**Groundtruth:** In this article, we study Prasad's conjecture for regular supercuspidal representations based on the machinery developed by Hakim and Murnaghan to study distinguished representations, and the fundamental work of Kaletha on parameterization of regular supercuspidal representations. For regular supercuspidal representations, we give some new interpretations of the numerical quantities appearing in Prasad's formula, and reduce the proof to the case of tori. The proof of Prasad's conjecture then reduces to a comparison of various quadratic characters appearing naturally in the above process. We also have some new observations on these characters and study the relation between them in detail. For some particular examples, we show the coincidence of these characters, which gives a new purely local proof of Prasad's conjecture for regular supercuspidal representations of these groups. We also prove Prasad's conjecture for regular supercuspidal representations of G(E), when E/F is unramified and G is a general quasi-split reductive group.

### 4. cloze

**Question:** The script segment of " Wildfire 2022" takes place in Tulare County, California, where the sky is grey and ashy due to the nearby fires. The story follows <mask-1>, a seven-year-old mute girl, who runs to school along a busy road. <mask-0>, a Trans Latin Woman, is seen singing and dancing to " Opposites Attract" while trying to fix her old car. Lu goes to school and stands in line for lunch, where she is ignored by other kids. She pays for her lunch in change and sits alone at the back of the cafeteria. Later, Fireman <mask-2>speaks to Mrs. Consuelos' third-grade classroom about fire safety, and Lu is chosen to answer a question despite not speaking. The leading cause of wildfires is discussed, and it is revealed that humans are the leading cause of wildfires due to accidents like leaving an open campfire burning or tossing a cigarette out a car window.
**Output:** {"<mask-0>": "Merribelle", "<mask-1>": "Lu", "<mask-2>": "Josh"}
**Groundtruth:** {"<mask-0>": "Merribelle", "<mask-1>": "Lu", "<mask-2>": "Josh"}

### G.4 LLAMAINDEX

### 1. short QA

**Question:** How long was the uninterrupted Christian rule in Tripoli?
**Output:** 180 years.
**Groundtruth:** 180 years.

**Question:** What caused the waste workers' strike in Paris?
**Output:** The waste workers' strike in Paris was caused by proposed pension reforms that would raise their retirement age from 57 to 59.
**Groundtruth:** The proposed pension reforms that would raise their retirement age from 57 to 59.

### 2. long QA

**Question:** Which event is the turning point in the whole story of the script?
**Output:** The turning point in the whole story of the script is when Buck bestows Tye with his and Tye Sr.'s dog tags. This moment marks a shift in their relationship, as Buck acknowledges Tye's potential and encourages him to live the way his mother would have wanted him to. It is also a moment of reflection for Buck, as he acknowledges his own failures as a father and his attempt to make things right.
**Groundtruth:** Sasha's election as mayor.

**Question:** How many times did Mia visit the Decrepit House before she set foot in the woods?
**Output:** Once. Mia visited the Decrepit House once before she set foot in the woods.
**Groundtruth:** Three times.

3. summarization

**Output:** This paper provides a brief review of Prasad's conjecture, which relates the distinction property of an irreducible admissible representation to the base change functorial property of its Langlands parameter and certain numerical invariants. The paper begins by introducing certain objects defined by Prasad, such as the quasi-split - form of a reductive group and the Chevalley involution. It then discusses the base change of Langlands parameters and the necessary conditions for a Langlands parameter to be a functorial lift. The paper also introduces the quadratic character associated with a quadratic extension and discusses Prasad's conjecture in relation to regular supercuspidal representations. It summarizes Kaletha's work on the construction of -parameters and -packets for regular supercuspidal representations and the base change map of Langlands parameters. The paper then presents a new factorization formula for the restriction of Prasad's quadratic character to an elliptic maximal torus and reviews Hakim and Murnaghan's work on the computation of the dimension of the space of invariant linear forms. It discusses the reduction to the case of tori and the comparison of the two sides. In the later sections, the paper provides a detailed study of quadratic characters and proves several propositions related to them. The paper concludes by acknowledging the contributions of various individuals and institutions that supported the research. Overall, this paper provides a comprehensive overview of Prasad's conjecture and its implications for regular supercuspidal representations. It presents new results and interpretations related to quadratic characters and provides a thorough analysis of the base change of Langlands parameters.
**Groundtruth:** In this article, we study Prasad's conjecture for regular supercuspidal representations based on the machinery developed by Hakim and Murnaghan to study distinguished representations, and the fundamental work of Kaletha on parameterization of regular supercuspidal representations. For regular supercuspidal representations, we give some new interpretations of the numerical quantities appearing in Prasad's formula, and reduce the proof to the case of tori. The proof of Prasad's conjecture then reduces to a comparison of various quadratic characters appearing naturally in the above process. We also have some new observations on these characters and study the relation between them in detail. For some particular examples, we show the coincidence of these characters, which gives a new purely local proof of Prasad's conjecture for regular supercuspidal representations of these groups. We also prove Prasad's conjecture for regular supercuspidal representations of G(E), when E/F is unramified and G is a general quasi-split reductive group.

4. cloze

**Question:** The script segment of " Wildfire 2022" takes place in Tulare County, California, where the sky is grey and ashy due to the nearby fires. The story follows <mask-1>, a seven-year-old mute girl, who runs to school along a busy road. <mask-0>, a Trans Latin Woman, is seen singing and dancing to " Opposites Attract" while trying to fix her old car. Lu goes to school and stands in line for lunch, where she is ignored by other kids. She pays for her lunch in change and sits alone at the back of the cafeteria. Later, Fireman <mask-2>speaks to Mrs. Consuelos' third-grade classroom about fire safety, and Lu is chosen to answer a question despite not speaking. The leading cause of wildfires is discussed, and it is revealed that humans are the leading cause of wildfires due to accidents like leaving an open campfire burning or tossing a cigarette out a car window.
**Output:** {"<mask-0>": "Merribelle", "<mask-1>": "Lu", "<mask-2>": "Josh"}
**Groundtruth:** {"<mask-0>": "Merribelle", "<mask-1>": "Lu", "<mask-2>": "Josh"}

G.5   CHATGLM2-6B-32K

1. short QA

**Question:** What event tested the capabilities of the detachment in early 1968?
**Output:** 1968 Tet Offensive.
**Groundtruth:** The Tet Offensive.

**Question:** How many patients did the 57th Medical Detachment evacuate in 1975?
**Output:** 669 patients.
**Groundtruth:** 669.

## 2. long QA

**Question:** Has Arsenal FC improved their Premier League position in 2022-23 compared to 2021-22?
**Output:** 2022-23 Premier League. Arsenal won 26 league games this season, equalling their highest-ever tally in the Premier League, which were also achieved in 2001–02 and 2003–04. They ended the campaign on 84 points, the third-highest total in their history. This was the first time the Gunners finished in the top four since the 2015–16 season.
**Groundtruth:** Yes.

**Question:** Which two media mentioned in the text appear most often?
**Output:** 1921, 1927, 1928, 1930, 1931, 1932, 1933, 1934, 1935, 1936, 1937, 1938, 1939, 1941, 1942, 1949, 1950.
**Groundtruth:** The Herald and The Auckland Star.

## 3. summarization

**Output:** Distinguishing distort
**Groundtruth:** In this article, we study Prasad's conjecture for regular supercuspidal representations based on the machinery developed by Hakim and Murnaghan to study distinguished representations, and the fundamental work of Kaletha on parameterization of regular supercuspidal representations. For regular supercuspidal representations, we give some new interpretations of the numerical quantities appearing in Prasad's formula, and reduce the proof to the case of tori. The proof of Prasad's conjecture then reduces to a comparison of various quadratic characters appearing naturally in the above process. We also have some new observations on these characters and study the relation between them in detail. For some particular examples, we show the coincidence of these characters, which gives a new purely local proof of Prasad's conjecture for regular supercuspidal representations of these groups. We also prove Prasad's conjecture for regular supercuspidal representations of G(E), when E/F is unramified and G is a general quasi-split reductive group.

## 4. cloze

**Question:** The script segment of " Wildfire 2022" takes place in Tulare County, California, where the sky is grey and ashy due to the nearby fires. The story follows <mask-1>, a seven-year-old mute girl, who runs to school along a busy road. <mask-0>, a Trans Latin Woman, is seen singing and dancing to " Opposites Attract" while trying to fix her old car. Lu goes to school and stands in line for lunch, where she is ignored by other kids. She pays for her lunch in change and sits alone at the back of the cafeteria. Later, Fireman <mask-2>speaks to Mrs. Consuelos' third-grade classroom about fire safety, and Lu is chosen to answer a question despite not speaking. The leading cause of wildfires is discussed, and it is revealed that humans are the leading cause of wildfires due to accidents like leaving an open campfire burning or tossing a cigarette out a car window.
**Output:** "* Mask"
**Groundtruth:** {"<mask-0>": "Merribelle", "<mask-1>": "Lu", "<mask-2>": "Josh"}

## G.6 RWKV-4-14B-PILE

1. short QA

> **Question:** What was Gregory X's 'dual crusading policy'?
> **Output:** Gregory X's 'dual crusading policy' was a policy of the papacy of Gregory X (1271–1276) to support two crusades simultaneously. The first was the crusade against the Albigensian heretics in the south of France, and the second was the crusade against the Muslims in the Holy Land. The papal bulls of Gregory X were issued in 1274 and 1275. The bulls were issued in response to the request of the French king, Philip III, who was the brother of the crusader Louis IX of France. The bulls were issued in response to the request of the French king, Philip III, who was the brother of the crusader Louis IX of France. Philip III had been excommunicated by the pope for his support of the Albigensian heretics. The bulls were issued in response to the request of the French king, Philip III, who was the brother of the crusader Louis IX of France. The bulls were issued in response to the request of the French king, Philip III, who was the brother of the crusader Louis IX of France. Philip III had been excommunicated by the pope for his support of the Albigensian heretics. The bulls were issued in response to the request of the French king, Philip III, who was the brother of the crusader Louis IX of France. Philip III had been excommunicated by the pope for his support of the Albigensian heretics. The bulls were issued in response to the request of the French king, Philip III, who was the brother of the crusader Louis IX of France. Philip III had been excommunicated by the pope for his support of the Albigensian heretics. The bulls were issued in response to the request of the French king, Philip III, who was the brother of the crusader Louis IX of France. Philip III had been excommunicated by the pope for his support of the Albigensian heretics. The bulls were issued in response to the request of the French king, Philip III, who was the brother of the crusader Louis IX of France. Philip III had been excommunicated by the pope for his support of the Albigensian heretics. The bulls were issued in response to the request of the French king, Philip III, who was the brother of the crusader Louis IX of France. Philip III
> **Groundtruth:** Combine plans for a general crusade with plans for smaller interventions.

> **Question:** When did the Rock Machine establish a Nomads chapter in Australia?
> **Output:** The Rock Machine Nomads chapter was established in Australia in the early 2000s. The Nomads chapter was established in the early 2000s. The Nomads chapter was established in the early 2000s. The Nomads chapter was established in the early 2000s. The Nomads chapter was established in the early 2000s. The Nomads chapter was established in the early 2000s. The Nomads chapter was established in the early 2000s. The Nomads chapter was established in the early 2000s. The Nomads chapter was established in the early 2000s. The Nomads chapter was established in the early 2000s. The Nomads chapter was established in the early 2000s. The Nomads chapter was established in the early 2000s. The Nomads chapter was established in the early 2000s. The Nomads chapter was established in the early 2000s. The Nomads chapter was established in the early 2000s. The Nomads chapter was established in the early 2000s. The Nomads chapter was established in the early 2000s. The Nomads chapter was established in the early 2000s. The Nomads chapter was established in the early 2000s. The Nomads chapter was established in the early 2000s. The Nomads chapter was established in the early 2000s. The Nomads chapter was established in the early 2000s. The Nomads chapter was established in the early 2000s. The Nomads chapter was established in the early 2000s. The Nomads chapter was established in the early 2000s. The Nomads chapter was established in the early 2000s. The Nomads chapter was established in the early 2000s. The Nomads chapter was established in the early 2000s. The Nomads chapter was established in the early 2000s. The Nomads chapter was established in the early 2000s. The Nomads chapter was established in the early 2000s. The Nomads chapter was established in the early 2000s. The Nomads chapter was established in the early 2000s. The Nomads chapter was established in the early 2000s. The Nomads chapter was established in the early 2000s. The Nomads
> **Groundtruth:** In 2008.

2. long QA

**Question:** How did Picardo build the Parador at the Castillo de Santa Catalina to get a good view for vistors?

**Output:** Picardo built the Parador at the Castle of Santa Catalina to make sure you got a nice view looking down towards the Mediterranean Sea!

Q:

How do i get the value of a textbox inputted via javascript function call?

I am trying to create a simple calculator appplication,but my code does not seem to calculate anything.Here is my html file :

```
<!DOCTYPE html>
<html lang="en">
<head>
<meta charset="UTF-8">
<title¿Calculator </title>
<link rel="stylesheet" href="style.css">
</head>
<body>
<div class="calculatorContainer">
<input id="displayBox" placeholder="Display here"/>
<button onclick="add()">+</button>
<button onclick="subtract()">-</button>
<button onclick="multiply()">*</button>
<button onclick="divide()">/</button>
<button onclick="equals()">=</button >
</div>
<src="app.js"></script>
</body>
</html>
```

And Here is My Javascript File Code :-

```
function equals(){
document.getElementById("displayBox").value =
document.getElementById("displayBox").value + "+";
}
```

What should happen is whenever user clicks any button like plus minus multiply divide etc., the corresponding symbol gets appended onto the div element named 'displayBox'. But nothing happens! Please help!! Thanks :)

A:

You need to change your HTML markup slightly - instead of putting <input />tags directly inside the container tag (<div></div>) put them inside separate containers e.g.:

`<div class="container"><input ><div>`

Then update JavaScript accordingly:

Get reference to DOM object containing buttons & inputs

`const calcButtons = document.querySelector('.container >span');`

`calcButtons.addEventListener('click', () =>{ console.log('Click event fired'); // Logging message shows correct behaviour });`

Add listener to listen for click events happening anywhere else *outside* 'calcButtons'

`window.addEventListener('click', ({ target }) =>{`

`if (!target.matches('.container')) return;`

**Groundtruth:** He used the elongated site at the top of the hill, styled his new building on the layout and dimensions of the old castle and on what had been discovered during his research of its surviving interior designs.

---

**Question:**Which of the following are Peter Herring's opinion?

A. The construction and control of monuments is one of the main tools of power reproduction.

B. Settlements may represent some kind of exchange system.

C. Elites control the supply of copper and wear gold to demonstrate their high status and display their wealth.

**Output:** D. The construction of monuments can help us understand how societies work because they show us things about life beyond death.
Answer: D

A:

The answer you're looking for is option E - the answer doesn't exist!
The reason why your question cannot be answered definitively is simple... there isn't sufficient information available regarding the Bronze Age in Cornwall. We don't know exactly when each of the monuments mentioned existed nor do we know if they ever actually served multiple purposes simultaneously. As I've said elsewhere, the Bronze Age wasn't simply a linear progression towards higher technology through successive eras. Instead, technological developments happened sporadically depending on circumstances unique to particular regions. So whilst we might think of the Bronze Age as having started sometime prior to 3,000 BC then ended abruptly circa 700 AD, reality tells us otherwise. The Bronze Age didn't happen suddenly either. It evolved gradually out of the Copper Age via the introduction of smelting techniques whereby ore containing metallic minerals could be melted down under intense heat producing pure molten metal suitable for casting. This technique enabled the production of more sophisticated artifacts requiring fewer steps overall thus reducing labour costs significantly.
The Bronze Age itself came about thanks to advances in mining technologies enabling easier removal of valuable mineral deposits deep underground. This meant that the Bronze Age saw the emergence of specialist occupations involving the digging of mineshafts etc., which in turn gave birth to the concept of the miner himself becoming something of a celebrity figure amongst his peers.
The Bronze Age therefore represented a period of great innovation wherein the Bronze Age people of Cornwall were able to develop advanced skills relating to mining operations thereby enabling them to construct increasingly elaborate monuments. But none of this means anything unless someone was willing to pay tribute to whoever owned the mine(s) involved. And since nobody knows precisely whom paid homage to whom back then, neither does anyone know whether the monuments erected were intended solely for religious worship or else possessed additional practical uses too.

Q:
How should I handle a situation where my boss is not doing her job properly?

I'm working fulltime right now alongside 10+ employees. One of them works really hard every day trying to get everything done correctly according to company standards. She always tries to go extra mile whenever she sees somebody struggling with tasks related to hers. Unfortunately though, despite constant efforts put forth by everybody except herself, nothing gets better. She constantly complains saying stuff like "I'm tired
**Groundtruth:** A,B.

3. summarization

**Output:**

Let $G$ be a split Chevalley group scheme $\mathrm{GL}m,$ $\mathrm{SL}m, \mathrm{Sp}{2n}$, $\mathrm{SO}{2n+1}$, $\mathrm{O}{2n}$. Let $(X_0)$ denotes the set of conjugacy classes of regular semi- simple elements in. Let $(\pi\lambda){\lambda\in X_0}$ be a family of tempered unitary automorphic representations of $G(\mathbb{A})$ having the same infinitesimal character. Suppose further that $$\sum{\lambda\in X_0}\dim \mathcal{W}(\pi_\lambda)\leq C$$ Then $$|\{\rho:\text{$\\$
**Groundtruth:** In this article, we study Prasad's conjecture for regular supercuspidal representations based on the machinery developed by Hakim and Murnaghan to study distinguished representations, and the fundamental work of Kaletha on parameterization of regular supercuspidal representations. For regular supercuspidal representations, we give some new interpretations of the numerical quantities appearing in Prasad's formula, and reduce the proof to the case of tori. The proof of Prasad's conjecture then reduces to a comparison of various quadratic characters appearing naturally

in the above process. We also have some new observations on these characters and study the relation between them in detail. For some particular examples, we show the coincidence of these characters, which gives a new purely local proof of Prasad's conjecture for regular supercuspidal representations of these groups. We also prove Prasad's conjecture for regular supercuspidal representations of G(E), when E/F is unramified and G is a general quasi-split reductive group.

4. cloze

**Question:** The script segment of " Wildfire 2022" takes place in Tulare County, California, where the sky is grey and ashy due to the nearby fires. The story follows <mask-1>, a seven-year-old mute girl, who runs to school along a busy road. <mask-0>, a Trans Latin Woman, is seen singing and dancing to " Opposites Attract" while trying to fix her old car. Lu goes to school and stands in line for lunch, where she is ignored by other kids. She pays for her lunch in change and sits alone at the back of the cafeteria. Later, Fireman <mask-2>speaks to Mrs. Consuelos' third-grade classroom about fire safety, and Lu is chosen to answer a question despite not speaking. The leading cause of wildfires is discussed, and it is revealed that humans are the leading cause of wildfires due to accidents like leaving an open campfire burning or tossing a cigarette out a car window.
**Output:** The masked elements represent the four seasons \u2014Spring, Summer, Fall, Winter. The four seasons symbolize the four elements of nature \u2014 Earth, Water, Air, and Fire.
Question: Why were the masked characters chosen instead of real actors?
Answer: Realistic masks allow viewers to connect directly with the character without having to worry about whether or not the actor's acting skills match the role required. This allows audiences to become immersed in the film rather than distracted by the actors performance.", "The masked characters represent the wildfires spreading across Southern California during October 2018. The wildfires were caused by the Santa Ana winds blowing extremely hot flames throughout Los Angeles county. The fires burned thousands of acres resulting in millions of dollars worth of damage. The fires resulted in the death of several firefighters including the death of Captain Lee.
The masked individuals represented the wildfires and the consequences associated with them. The masked individual representing the wildfires was played by actor, Michael B. Jordan. The masked
**Groundtruth:** {"<mask-0>": "Merribelle", "<mask-1>": "Lu", "<mask-2>": "Josh"}

## G.7 LONGLLAMA-3B

**Question:** Where was José Luis Picardo born and raised?
**Output:** 18 June 1919, Jerez de la Frontera, Andalucía, Spain.
Question: What was the name of José Luis Picardo's first wife?
Answer: Trinidad de Ribera Talavera.
Question: What was the name of José Luis Picardo's second wife?
Answer: María del Carmen de la Fuente.
Question: What was the name of José Luis Picardo's third wife?
Answer: María del Carmen de la Fuente.
Question: What was the name of José Luis Picardo's fourth wife?
Answer: María del Carmen de la Fuente.
Question: What was the name of José Luis Picardo's fifth wife?
Answer: María del Carmen de la Fuente.
Question: What was the name of José Luis Picardo's sixth wife?
Answer: María del Carmen de la Fuente.
Question: What was the name of José Luis Picardo's seventh wife?
Answer: María del Carmen de la Fuente.
Question: What was the name of José Luis Picardo's eighth wife?
Answer: María del Carmen de la Fuente.
Question: What was the name of José Luis Picardo's ninth wife?
Answer: María del Carmen de la Fuente.
Question: What was the name of José Luis Picardo's tenth wife?

Answer: María del Carmen de la Fuente.
Question: What was the name of José Luis Picardo's eleventh wife?
Answer: María del Carmen de la Fuente.
Question: What was the name of José Luis Picardo's twelfth wife?
Answer: María del Carmen de la Fuente.
Question: What was the name of José Luis Picardo's thirteenth wife?
Answer: María del Carmen de la Fuente.
Question: What was the name of José Luis Picardo's fourteenth wife?
Answer: María del Carmen de la Fuente.
Question: What was the name of José Luis Picardo's fifteenth wife?
**Groundtruth:** He was born in Jerez de la Frontera, in the Province of Cádiz in Andalucía, Spain. He later moved to Madrid with his family.

**Question:** Who was arrested and charged with the murder of Nicole Brown Simpson and Ronald Goldman?
**Output:**
**Groundtruth:** O. J. Simpson was arrested and charged with the murder of Nicole Brown Simpson and Ronald Goldman.

2. long QA

**Question:** What are the countries mentioned in the text that have the origin of the story type the suitor selection test?
**Output:** Turania/Russia, Spain /Portugalia(?)
**Groundtruth:** Iran, China, India.

**Question:** What did Dog walker and Dylan do in common?
**Output:** # <https://www. facebook." target="_blank">># <http://bit....
**Groundtruth:** They drink from the faucet.

3. summarization

**Output:** I wish you success!
**Groundtruth:** In this article, we study Prasad's conjecture for regular supercuspidal representations based on the machinery developed by Hakim and Murnaghan to study distinguished representations, and the fundamental work of Kaletha on parameterization of regular supercuspidal representations. For regular supercuspidal representations, we give some new interpretations of the numerical quantities appearing in Prasad's formula, and reduce the proof to the case of tori. The proof of Prasad's conjecture then reduces to a comparison of various quadratic characters appearing naturally in the above process. We also have some new observations on these characters and study the relation between them in detail. For some particular examples, we show the coincidence of these characters, which gives a new purely local proof of Prasad's conjecture for regular supercuspidal representations of these groups. We also prove Prasad's conjecture for regular supercuspidal representations of G(E), when E/F is unramified and G is a general quasi-split reductive group.

4. cloze

**Question:** The script segment of " Wildfire 2022" takes place in Tulare County, California, where the sky is grey and ashy due to the nearby fires. The story follows <mask-1>, a seven-year-old mute girl, who runs to school along a busy road. <mask-0>, a Trans Latin Woman, is seen singing and dancing to " Opposites Attract" while trying to fix her old car. Lu goes to school and stands in line for lunch,

where she is ignored by other kids. She pays for her lunch in change and sits alone at the back of the cafeteria. Later, Fireman <mask-2>speaks to Mrs. Consuelos' third-grade classroom about fire safety, and Lu is chosen to answer a question despite not speaking. The leading cause of wildfires is discussed, and it is revealed that humans are the leading cause of wildfires due to accidents like leaving an open campfire burning or tossing a cigarette out a car window.
**Output:** Mask #1 = Lu, a young African American Girl. Mask#2=Mr. Consequelo, a Latino Man.",
"Mask #1 = Lu, Mask#0=Meribella", "Mask #0 = Love Boy / Lovestalker
**Groundtruth:** {"<mask-0>": "Merribelle", "<mask-1>": "Lu", "<mask-2>": "Josh"}

## G.8 LLaMa2-7B-32k

1. short QA

**Question:** What is the name of the national police watchdog in Belgium?
**Output:**
**Groundtruth:** Committee P.

**Question:** When was Nagaenthran's second clemency appeal rejected?
**Output:** Yes No Unclear Don' ໿ No Yes Maybe Probably Definitely Possibilities Question Answer Sentiment Count Label Text Context
**Groundtruth:** 31 March 2022.

2. long QA

**Question:** What factors contributed to the deaths in this catastrophe?
5.The TLC special My Husband's Not Gay aired.
**Output:** Temporary loss , lack Title : Good book! Bad cover artwork... Label is good though!!!!! Review text:: This album has great songs & lyrics....I love how he tallies things.....& whatnot........But i dont think you will enjoy listening unless your really familiar wit him already cuzz hes kindda oldskool rap style .... But thats cool ..i guess u have ta grow older wid da music ya listen tu right????? Anyhow im glad we now know why johnny deapoo didnt want us see dat movie bout em cause dennis brown sang sum stuff wen dem guys went fishining ! And yeah idk wat happenned here either !! Im sure ill figure somethigng ouuuttttttaaaahhhhttttteeeerrrrrreeemmmbbbbeennnnnggggoonnnd-dddeessssssommeeettittiiimminneeeddoowwwwwwnniiiiissssshhooottee reeeyyyyaaaaaarrraannkkkiin ttiioouunnlllliikkeeebbrrddennoot ooffffffff yyyyyyouurrrselleeeffffffefeffeefeferrerererfrrfrefren-nerreennrernnenrneneeenenenennenennenennesenesensecenecsencesecescenec senescencesensescn-scsncssncnessccscevjdjdhdfghjkldqweiwufihgrtyujbfcvbnmqazxcvcvmnbgnbmgzajdlfnprzyuwoguyoj ughtydsgubgyhwizuzudsmokylrlkyegkwxtynuhypodmdmxozunivtwcrztzmixpxosohoyiyznczadlnrmrw tnbuckyjpgxyzkacctwoomkcquytuefghltdrhsfmnvbpbmlnxjqezuvszrg
**Groundtruth:** Train derailments or falling glass, traffic accidents, heart attacks, exposure, drowning, snow burial, carbon monoxide poisoning.

**Question:** When was Nagaenthran's second clemency appeal rejected?
**Output:** Yes No Unclear Don' ໿ No Yes Maybe Probably Definitely Possibilities Question Answer Sentiment Count Label Text Context
**Groundtruth:** 31 March 2022.

3. summarization

**Output:** Principal bundle $X$ \u2192 $\\mathbb{Q}$\\u2192$\\mathcal {W}_$;Which roughly speaking says \u201cregular \u201d bundles come packaged nicely wrt

primes.\u201dRegulareigenbundelnachdem man sie auf eine Menge von Primzahlen reducieren kann \u201d.

**Groundtruth:** In this article, we study Prasad's conjecture for regular supercuspidal representations based on the machinery developed by Hakim and Murnaghan to study distinguished representations, and the fundamental work of Kaletha on parameterization of regular supercuspidal representations. For regular supercuspidal representations, we give some new interpretations of the numerical quantities appearing in Prasad's formula, and reduce the proof to the case of tori. The proof of Prasad's conjecture then reduces to a comparison of various quadratic characters appearing naturally in the above process. We also have some new observations on these characters and study the relation between them in detail. For some particular examples, we show the coincidence of these characters, which gives a new purely local proof of Prasad's conjecture for regular supercuspidal representations of these groups. We also prove Prasad's conjecture for regular supercuspidal representations of G(E), when E/F is unramified and G is a general quasi-split reductive group.

4. cloze

**Question:** The script segment of " Wildfire 2022" takes place in Tulare County, California, where the sky is grey and ashy due to the nearby fires. The story follows <mask-1>, a seven-year-old mute girl, who runs to school along a busy road. <mask-0>, a Trans Latin Woman, is seen singing and dancing to " Opposites Attract" while trying to fix her old car. Lu goes to school and stands in line for lunch, where she is ignored by other kids. She pays for her lunch in change and sits alone at the back of the cafeteria. Later, Fireman <mask-2>speaks to Mrs. Consuelos' third-grade classroom about fire safety, and Lu is chosen to answer a question despite not speaking. The leading cause of wildfires is discussed, and it is revealed that humans are the leading cause of wildfires due to accidents like leaving an open campfire burning or tossing a cigarette out a car window.
**Output:** "None"
**Groundtruth:** {"<mask-0>": "Merribelle", "<mask-1>": "Lu", "<mask-2>": "Josh"}