# OpenReview forum: "Can long-context large language models understand long contexts?"
_ICLR.cc/2024/Conference — ICLR 2024 Conference Withdrawn Submission_

### Official Review · Reviewer_8ZV2 · 2023-10-17

**Soundness:** 2 fair
**Presentation:** 1 poor
**Contribution:** 3 good
**Rating:** 3
**Confidence:** 4

**Summary:**

The paper presents a new benchmark called LooGLE for long-context LLMs, with inputs longer than 24k tokes. Some/most examples are human-annotated, and some/most were cross-validated by multiple annotators.
Further, some/most of the documents in the benchmark were published after 2022, which is expected to be after the knowledge cutoff of LLMs such as GPT-3.5 and GPT-4, forcing them to rely only on their in-context learning abilities rather than their prior knowledge.

Such benchmarks are always useful and needed in the community, especially following the growing interest in long-context LLMs.

**Strengths:**

* The authors made lots of efforts in collecting and curating the data.
* Such benchmarks are always useful and needed in the community, especially following the growing interest in long-context LLMs.

**Weaknesses:**

- Following the promises in the abstract about the human-annotation and the cross-annotator validation, I was very disappointed to see that a large part of the benchmark's ground-truth output was generated using GPT 3.5 / 4:
>We utilize the powerful language processing and understanding capability of GPT-3.5-turbo to help generating short QA pairs from the original text.

>we employ GPT-3.5-turbo to generate factual summaries align with the source segment using with constraints

If this is indeed the case, this is disappointing, and the benchmark may be biased toward "questions that are easy for ChatGPT to answer".

- The text is very unclear in many cases. For example when describing the statistics of the benchmark, the paper says:
>Extra-long realistic documents. It contains 778 latest gathered and extremely long documents
with an average of 16.4k words. There are over 6000 test instances without distribution bias for a
more generalized assessment, many of which are exceeding 100k words.

So are there 778 examples or 6000 examples? If "many of which exceed 100k words", how many of them? what's the average? Are these two datasets? If not, why are these numbers reported separately?

- The results in Section 4.3.1 are very confusing and unclear. For example:
>In Table 3, it can be noticed that LlamaIndex obtains from the perspective of GPT4 evalution. Instead
of memorizing a shortcut of original input with a limited context window, retrieval-based context
compression technique augments the LLM by incorporating external memory, allowing relevant
information to be retrieved using a specific query.

I am not sure what such paragraphs are trying to say. What does it mean that "LlamaIndex obtains from the perspective of GPT4 evalution"? What do the authors exactly mean by "memorizing a shortcut"? Who is memorizing a shortcut?
- Measuring "GPT4 score" on GPT4's outputs is mostly meaningless. It would be better to just completely remove this column, or use another LLM that is not evaluated.
- Applicability: the paper does not mention anything about its implementation, its ease of use, its availability. As always with benchmarks, the devil is in the details, and the authors have not included the data itself, which makes it hard to really evaluate its quality.
- Presentation is poor: for example:
    - the text in Figure 1 is tiny, not allowing to actually understand the overview of the new benchmark.
The entire left part of the figure contains barely any information.
I would prefer an organized and readable list of tasks and data statistics.

    - The text in Table 1 tiny
    - The text in Table 2 is tiny. Further, it would be helpful if these statistics would include the max/min instead of category, or a more illustrative figure of the characteristics of the examples, as in Figure 1 in the [SCROLLS paper](https://arxiv.org/pdf/2201.03533.pdf)
    - The text in Figure 3 is tiny. Further, the colors are very similar, and I cannot distinguish between the different models and cannot understand anything from this figure.

**Questions:**

### Questions
1. Section 3.3.1 says that "we directly use the abstract of each paper as the reference for generating summaries" - so, the ground-truth summaries where **generated**? are the Abstracts **used** in any part of the process other than for evaluation?
2. Are the authors going to release the test sets, or keep them "hidden"?
3. Are there training/test spits, or is everything "test"?

### Comments
1. The comparison in Table 1 on "which tasks are included in each benchmark" shows that LooGLE contains many tasks that other prior benchmarks do not. However, it is a bit unfair, because these prior benchmarks contain tasks that are not contained in LooGLE, but these are not mentioned. For example, Scrolls, mentioned in the first line, does contain QA (mentioned with "X") and NLI (not mentioned at all).

### Summary
I appreciate the authors's efforts, but as much as good benchmarks are needed in the community, unfinished benchmarks can do harm and drive research in the wrong direction.
I cannot evaluate the benchmark itself since it was not released, but the paper still feels a bit unclear and unfinished, which makes me worry that the benchmark is too.
Thus, I currently vote for rejection, and hope that the authors would polish both the paper and the benchmark and release them when they are in a more polished state.

---

> ### Author Response · Authors · 2023-11-16
> **To Reviewer 8ZV2**
>
> > Q1: Following the promises in the abstract about the human-annotation and the cross-annotator validation, I was very disappointed to see that a large part of the benchmark's ground-truth output was generated using GPT 3.5 / 4. If this is indeed the case, this is disappointing, and the benchmark may be biased toward "questions that are easy for ChatGPT to answer".
>
> **A:** We thank the reviewer for pointing out this concern. We indeed have spent a huge effort and high cost to make the evaluation as fair as possible in the following steps:
> - **For short dependency QA**, we have manually reviewed all the short QA pairs and carefully refined the answers to make them clear and concise. The model is forced to extract the answer directly from the original document to generate the initial answer. Then we make refinements by filtering out the non-essential contexts and removing redundant descriptions from the model.
> - **For long dependency QA**, there are over 1100 long dependency QA pairs delicately designed by human annotators, despite the high costs and huge effort. Each document for generating QA pairs underwent a meticulous three-step process(Question & answer, Answer & check, Cross-validation & revise) that involved the assignment of two distinct annotators who are unaware of each other’s identities. The annotation adhered to stringent standards including long dependency, diverse problem types, clear & precise questions and deterministic & objective answers. Participants were prohibited from using large language models and tools like ChatGPT for article reading, data generation, and annotation.
> 3) **For summarization**, answers are abstracts extracted from the paper.
> 4) **For cloze**, we did not use gpt-3.5 to generate both the question and answer directly, and the answers are extracted by a NER model.
>
> **This rigorous curation process was undertaken to ensure the high quality of the questions, answers, as well as supporting evidence. This approach aims to achieve questions with a high degree of accuracy, precision, and relevance to the document’s content.**
>
>
> > Q2: The text is very unclear in many cases. For example when describing the statistics of the benchmark, the paper says: Extra-long realistic documents. It contains 778 latest gathered and extremely long documents with an average of 16.4k words. There are over 6000 test instances without distribution bias for a more generalized assessment, many of which are exceeding 100k words. So are there 778 examples or 6000 examples? If "many of which exceed 100k words", how many of them? what's the average? Are these two datasets? If not, why are these numbers reported separately?
>
> **A:** Thanks for pointing out the misleading data statistics. For the polished version of the dataset, we have collected 776 original long documents, 2 of which exceed 100k words while 30% of which exceed 20k. The average words of LooGLE is more than 2 times longer than the existing dataset for long context. Please refer to **Figure 4** for the data length distribution. Based on the documents, we further generate over 6000 questions for test and evaluation. Details and numbers of the questions in each task, and their distributions, can be found in Table 2. We have also revised the expressions in the paper to make it precise.
>
> > Q3: The results in Section 4.3.1 are very confusing and unclear. For example: In Table 3, it can be noticed that LlamaIndex obtains from the perspective of GPT4 evalution. Instead of memorizing a shortcut of original input with a limited context window, retrieval-based context compression technique augments the LLM by incorporating external memory, allowing relevant information to be retrieved using a specific query. I am not sure what such paragraphs are trying to say. What does it mean that "LlamaIndex obtains from the perspective of GPT4 evalution"? What do the authors exactly mean by "memorizing a shortcut"? Who is memorizing a shortcut?
>
> **A:** Thank you for your suggestion. We intend to show that the retrieval-based LlamaIndex with external memory does help short QA to some extent and has a competitive performance compared to the winning GPT models, as inferred from Table 3. LLMs inherently lose much information in long inputs with limited context windows (which we call "memorizing a shortcut"). We have rewritten this part and hope that the edited section clarifies these key findings.

---

> > ### Author Response · Authors · 2023-11-16
> > **To Reviewer 8ZV2**
> >
> > > Q4: Measuring "GPT4 score" on GPT4's outputs is mostly meaningless. It would be better to just completely remove this column, or use another LLM that is not evaluated.
> >
> > **A:** Thanks for pointing out. We have indeed taken this into consideration and spared no efforts to make the evaluation as fair as possible in the following steps:
> > - **For evaluation**,
> > 1) There have been **many recent research studies that have shown that the GPT4 evaluator exhibits high consistency with human evaluation and can serve as a reliable annotator** to some extent. Here are some related works for your reference.
> > [1] Judging LLM-as-a-Judge with MT-Bench and Chatbot Arena
> > [2] Do large language models show decision heuristics similar to humans? a case study using gpt-3.5
> > [3] Calibrating llm-based evaluator
> > 3) We randomly selected over 400 questions from each task in long dependency QA and evaluated the accuracy from both GPT4's and the human perspective. The accuracy (%) evaluated by both methods and their agreement (%) are as follows:
> >     Method |Comprehension & reasoning|Computation | Timeline reorder | Information retrieval| Average
> >     ---|---|----|-----|----|-----
> >     Human evaluation| 51 | 21|26|29| 36
> >     GPT4 evaluation|53| 29 | 25| 34| 38
> >     Agreement|77| 89 | 76| 80| 80
> >
> >     **It can be seen that GPT4 can function well to make human-like judgement in our dataset**. In order to provide a more comprehensive assessment, we utilize GPT4 as an LLM evaluator to obtain reference results.
> > 3) Besides, **we make our implementation reproducible and GPT4's judgment deterministic** by setting its temperature to 0, top_p to 1, and prompting GPT4 to output True/False/exact-score only, instead of descriptive results. From our observations in the experiment results, we found that the GPT4 evaluator has no bias in itself when scoring.
> > 4) **How we use GPT4-eval is also delicately designed to ensure fairness**. By giving the question(QA only), groundtruth and predicted outputs, we ask GPT4 to compare and score considering the semantic matching, information completeness, consistency, fluency, and grammar. In this way, GPT4 can focus on the comparisons without bias and tendency for better evaluation. The detailed prompt can be seen in Appendix E.7 and E.8.
> >
> > - **For data generation**,
> > We indeed make a huge effort to avoid bias and keep fairness in evaluation:
> > 1) **For short QA**, we have manually reviewed all the short QA pairs and carefully refined the answers to make them clear and concise. The model is forced to extract the answer directly from the original document to generate the inital answer. Then we make refinements by filtering out the non-essential contexts and removing redundant descriptions from the model.
> > 2) **For long QA**, there are over 1100 long dependency QA pairs delicately designed by human annotators, despite the high costs and huge effort. Each document for generating QA pairs underwent a meticulous three-step process (Question & Answer, Answer & Check, Cross-Validation & Revise) that involved the assignment of two distinct annotators, who were unaware of each other's identities. The annotation adhered to stringent standards including long dependency, diverse problem types, clear & precise questions and deterministic & objective answers. Participants were prohibited from using large language models and tools like ChatGPT for article reading, data generation, and annotation.
> > 3) **For summarization**, answers are abstracts extracted from the paper.
> > 4) **For cloze**, we did not use gpt-3.5 to generate both the question and answer directly, and the answers are extracted by a NER model.
> >
> > **This rigorous curation process was undertaken to ensure the high quality of the questions, answers, as well as supporting evidence**. It also benefits from eliminating the interference from different model output formats, lengths, and other characteristics as much as possible.

---

> > > ### Author Response · Authors · 2023-11-16
> > > **To Reviewer 8ZV2**
> > >
> > > > Q5: Applicability: the paper does not mention anything about its implementation, its ease of use, its availability. As always with benchmarks, the devil is in the details, and the authors have not included the data itself, which makes it hard to really evaluate its quality.
> > >
> > > **A:** Thanks for pointing out your concern.Actually, we have already discussed the implementation of the dataset, from data collection and task definition to generation, especially the manual compilation of long dependency QA, in Section 3, which spans three pages. As for the data release and availability, our polished dataset has already been released on the HuggingFace Datasets and can be easily downloaded anytime for use. Meanwhile, we provide a brief summary and detailed instructions on the acquisition and usage of the dataset on the main page of LooGLE as well as on GitHub for better use. But to meet the anonymity requirement from the conference, we couldn't make it public in this paper and hope for your understanding. We will continuously and periodically update the dataset with newer documents, diverse and challenging tasks, as well as assessments for the latest long context LLMs, possibly on a yearly basis. Besides, our data collection process is fully open-sourced, making it easy for the community to reconstruct/update the benchmark.
> > >
> > > > Q6: Presentation is poor: for example:
> > > the text in Figure 1 is tiny, not allowing to actually understand the overview of the new benchmark. The entire left part of the figure contains barely any information. I would prefer an organized and readable list of tasks and data statistics.
> > > The text in Table 1 tiny
> > > The text in Table 2 is tiny. Further, it would be helpful if these statistics would include the max/min instead of category, or a more illustrative figure of the characteristics of the examples, as in Figure 1 in the SCROLLS paper
> > > The text in Figure 3 is tiny. Further, the colors are very similar, and I cannot distinguish between the different models and cannot understand anything from this figure.
> > >
> > > **A:** Thanks for your suggestion.
> > > - We have redrawn Figure 1, providing a clear representation of each task along with examples to aid readers in understanding each task.
> > > - We have refined Tables 1 and 2. Also, we have added the max/min length of the document data in Table 2.
> > > - For Figure 3, we have replaced it with a enhanced version of high resolution for better illustration with distinction.
> > > - For more information about the data statistics please refer to the **Appendix A**.
> > >     1. We have counted the length distribution of the dataset; please refer to **Figure 4**.
> > >     2. We also have counted the length of evidence spans for all the long dependency questions, please refer to **Figure 5**.
> > >         Dataset |Max.words|Min.words |
> > >         ---|---|----|
> > >         arXiv papers| 197,977 | 10,204|
> > >         Wikipedia pages|46,250| 11,285 |
> > >         Movie and TV scripts|62,751| 11,089 |
> > >
> > > > Q7: Section 3.3.1 says that "we directly use the abstract of each paper as the reference for generating summaries" - so, the ground-truth summaries where generated? are the Abstracts used in any part of the process other than for evaluation?
> > >
> > > **A:** Thank you for your thoughtful feedback. We use the original abstract directly from each paper without generating it ourselves, to make it the golden truth. The abstract is not used in any part of the process, other than the evaluation, to make the evaluation fair. We have also added relevant explanations in Section 3.2.1 in the implementation of paper summarization.
> > >
> > > > Q8: Are the authors going to release the test sets, or keep them "hidden"? Are there training/test spits, or is everything "test"?
> > >
> > > **A:** Currently, our polished dataset has been released in the HuggingFace Datasets as the dev set with all the golden truths. In the future, we will keep the dataset updated with newer documents and diverse tasks with more examples split for different sets and uses.

---

> > > > ### Author Response · Authors · 2023-11-16
> > > > **To Reviewer 8ZV2**
> > > >
> > > > > Comments: The comparison in Table 1 on "which tasks are included in each benchmark" shows that LooGLE contains many tasks that other prior benchmarks do not. However, it is a bit unfair, because these prior benchmarks contain tasks that are not contained in LooGLE, but these are not mentioned. For example, Scrolls, mentioned in the first line, does contain QA (mentioned with "X") and NLI (not mentioned at all).
> > > >
> > > > **Response to Comments**:
> > > > Thank you for your suggestion. As a distinctive novelty in our dataset, we place more attention on long dependency tasks and highlight them in Table 1. Most tasks in the existing multitask and multilingual dataset can be well-performed by LLMs when provided with longer inputs and less information loss. Tasks in LooGLE call for further comprehension and modeling of the interdependency among widely spread segments in the whole inputs, and further retrieval, computation, and reasoning based on that. Meanwhile, we will continually collect updated content and explore more challenging, multiple tasks to enhance LooGLE in the future.
> > > >
> > > >
> > > > > Summary: I appreciate the authors's efforts, but as much as good benchmarks are needed in the community, unfinished benchmarks can do harm and drive research in the wrong direction. I cannot evaluate the benchmark itself since it was not released, but the paper still feels a bit unclear and unfinished, which makes me worry that the benchmark is too. Thus, I currently vote for rejection, and hope that the authors would polish both the paper and the benchmark and release them when they are in a more polished state.
> > > >
> > > > **Response to Summary**:  We sincerely thank the reviewer for their considerable suggestions and we totally agree with them. We have indeed spent extremely high costs and a huge effort in the entire process of data collection & selection, task generation, and comprehensive evaluation. For the over 1100 long QA pairs, they were delicately designed and generated by human annotators from the beginning. Each QA pair underwent a meticulous three-round process with intensive cross-validation and revisions. The annotation adhered to stringent standards, including long dependencies, diverse problem types, clear, and precise questions. Deterministic and objective answers are prohibited from using tools like LLM. For short QA and clozes, we have manually and continuously done much data processing work for refinement. It takes nearly 4 months of the whole process to get all the data annotated, verified and polished for public. We believe that the current version is already clear and complete for release.
> > > > We believe that both of our goals are to benefit the community with more high-quality and useful benchmarks.

---

> > > > > ### Comment · Reviewer_8ZV2 · 2023-11-20
> > > > > **Response to authors**
> > > > >
> > > > > Thank you for your response.
> > > > >
> > > > > >The model is forced to ...
> > > > >
> > > > > How do you "force" the model?
> > > > >
> > > > > >There have been many recent research studies that have shown that the GPT4 evaluator exhibits high consistency with human evaluation and can serve as a reliable annotator to some extent.
> > > > >
> > > > > Perhaps, but when GPT-4 evaluates **its own predictions**, I do not think that it is a reliable annotator.
> > > > >
> > > > > Further, it might be an OK compromise to use GPT-4 as an evaluator in tasks that are hard to evaluate automatically. However when creating a new benchmark, I expect it to be perfectly accurate and clean. As shown in the authors' table, the agreement between GPT-4 and human evaluation is only around 80%.
> > > > >
> > > > > >How we use GPT4-eval is also delicately designed to ensure fairness. By giving the question(QA only), groundtruth and predicted outputs, we ask GPT4 to compare and score considering the semantic matching, information completeness, consistency, fluency, and grammar. In this way, GPT4 can focus on the comparisons without bias and tendency for better evaluation.
> > > > >
> > > > > I don't think that anything that GPT-4 is asked to do can be referred to as "ensured". How can the authors claim that this "*ensures* fairness" and "without bias"?
> > > > >
> > > > > >our polished dataset has already been released on the HuggingFace Datasets and can be easily downloaded anytime for use
> > > > >
> > > > > I'm happy to hear that the dataset is publicly available already, but I'm not sure how can I evaluate the dataset's quality without breaking anonymity.
> > > > >
> > > > > >In the future, we will keep the dataset updated with newer documents and diverse tasks with more examples split for different sets and uses.
> > > > >
> > > > > I actually think that datasets should better remain completely *frozen* rather than continuously updated.

---

### Official Review · Reviewer_oc4r · 2023-10-31

**Soundness:** 2 fair
**Presentation:** 1 poor
**Contribution:** 2 fair
**Rating:** 3
**Confidence:** 4

**Summary:**

This paper presents a new benchmark for long context understanding, consisting of 24k tokens in average. Firstly, this has the advantage to be more challenging than former benchmarks that have shorter texts compared with current LLMs' context window length (reaching up to 32k tokens). Secondly, it only contains newly created documents (after 2022) which are thus not present in most LLMs' pretraining data, preventing data leakage and enabling fairer evaluation. Experiments on current state-of-the-art LLMs reveal challenges in long-context understanding.

**Strengths:**

- A curated dataset, thoughtfully designed with great efforts to prevent data leakage and ensure long dependency.
- Great efforts in assessing current state-of-the-art LLMs' long dependency capabilities such as information retrieval, reading comprehension and reasoning, computation and timeline reordering.

**Weaknesses:**

- see questions.
- the paper is sprinkled with typos (refer to questions for a few).

**Questions:**

- Is the benchmark english-only ? If so, this needs to be mentioned.
- How will the dataset be released ? For instance ZeroSCROLLS only released the inputs and evaluate through a system of leaderboard, will LooGLE be released the same way ? I'm concerned that revealing input/gold outputs pairs would lead to data leakage for future models.
- Concerning data collection, were the sourced documents (after 2022) subjected to any machine-generated verification ? I am concerned that ChatGPT-like texts might compromise the fairness of the evaluation.
- Are the open-source models instruction-tuned ? Commercial closed source models like GPT3.5 rely on RLHF or some instruction tuning techniques that enable them to better follow instructions. If the considered open-source baselines are not instruction-tuned, the comparison might be unfair since the prompt used for evaluation is the same and is instruction-based.
- Is it fair to have GPT4 both as baseline and evaluator ? I am also concerned about the creation of the dataset of short QAs (generated by GPT3.5). Is it fair to evaluate a model that was used to create the dataset ?
- In section 4.2, you mention human evaluation (3) but I cannot find any human evaluation both in the paper and the appendix.
- Section 4.3.2 is confusing, the experiments are based on the recent work from Liu et al. 2023, that accessing information in the middle of the document is more challenging for LLMs. If I understood correctly, you suggest concatenating the head and the tail of the inputs and give it to the LLMs as input. This would mean discarding the whole "middle" and leaving the beginning and the end. This puts an immediate limitation on information that can be retrieved by the LLMs. Also did you only use the same model (GPT4-32k) but with various context input length or different variants of GPT4 ? What is the last entry in Table 5 ? (GPT4). For long summarization, arxiv abstracts are highly biased towards the beginning of the article so it is expected that increasing context would result in a higher divergence between the generated summary (which will contain more and more details) and the (gold) abstract.

**Typo**
- page 2: "Mannually designed both short and long dependency tasks"
- page 3: "COmputation."
- page 7: "Retrival"
- page 8: "GPT4 evalution"
- table 4: "Performence"
- appendix page 3: "Dispcrepancy"
- not really typos but it is uninformative to report results of the order of "e-300" since at this point there is nothing to really compare.

---

> ### Author Response · Authors · 2023-11-16
> **To Reviewer oc4r**
>
> > Q1: Is the benchmark english-only ? If so, this needs to be mentioned.
>
> **A:** The current version of the dataset is English-only. Thanks for your reminder and we have included this detail in the paper as well.
>
> > Q2: How will the dataset be released ? For instance ZeroSCROLLS only released the inputs and evaluate through a system of leaderboard, will LooGLE be released the same way ? I'm concerned that revealing input/gold outputs pairs would lead to data leakage for future models.
>
> **A:** Thanks for your comments and it's a great suggestion.Currently, our polished dataset has been released in the HuggingFace Datasets as the dev set, with all the golden truths. In the future, we will keep updating the dataset with newer documents and diverse tasks, providing more examples split for different sets and uses.
>
>
> > Q3: Concerning data collection, were the sourced documents (after 2022) subjected to any machine-generated verification ? I am concerned that ChatGPT-like texts might compromise the fairness of the evaluation.
>
> **A:** We thank the reviewer for your useful comments For data collection and selection, the sources of original documents (arXiv papers, Wikipedia articles, Movie and TV scripts) are most officially used and widely acknowledged to ensure their high quality. We will keep the data updated and follow your comments on data collection to guarantee the high quality of the dataset.
>
>
> > Q4: Are the open-source models instruction-tuned ? Commercial closed source models like GPT3.5 rely on RLHF or some instruction tuning techniques that enable them to better follow instructions. If the considered open-source baselines are not instruction-tuned, the comparison might be unfair since the prompt used for evaluation is the same and is instruction-based.
>
> **A:** Thank you for the valuable suggestion and we totally agree with that.
> Among the four open source models we used in this paper, **ChatGLM2-6B** was trained with human preference alignment and **LLaMA2-7B-32K** was instruction tuned for summarization and long context QA. We have highlighted this information in Section 4.1 in the revised paper. Based on your suggestion, we are working on additional experiments for the instruction tuning versions of the other two models. The results for long dependency QA can be seen below:
> | Models         | Context | Bleu1 | Bleu4 | Rouge1 | Rouge4 | RougeL | Meteor_score | Bert_score | GPT4 evaluator |
> |----------------|---------|-------|-------|--------|--------|--------|--------------|------------|----------------|
> | Long_llama-3b-instruct  | 256k     | 5.64  | 0.49  | 17.30  | 3.76   | 16.29  | 6.53         | 84.26      | 21.64          |
> | RWKV-4-raven-14b | 8k     | 3.88  | 0.22  | 20.39  | 3.20   | 19.20  | 6.41         | 81.46      | 14.32         |

---

> > ### Author Response · Authors · 2023-11-16
> > **To Reviewer oc4r**
> >
> > > Q5: Is it fair to have GPT4 both as baseline and evaluator ? I am also concerned about the creation of the dataset of short QAs (generated by GPT3.5). Is it fair to evaluate a model that was used to create the dataset ?
> >
> > **A:** Thanks for the insightful question. We have indeed taken this into consideration and spared no efforts to make the evaluation as fair as possible in the following steps:
> > - **For evaluation**,
> > 1) There have been **many recent research studies that have shown that the GPT4 evaluator exhibits high consistency with human evaluation and can serve as a reliable annotator** to some extent. Here are some related works for your reference.
> > [1] Judging LLM-as-a-Judge with MT-Bench and Chatbot Arena
> > [2] Do large language models show decision heuristics similar to humans? a case study using gpt-3.5
> > [3] Calibrating llm-based evaluator
> > 3) We randomly selected over 400 questions from each task in long dependency QA and evaluated the accuracy from both GPT4's and the human perspective. The accuracy (%) evaluated by both methods and their agreement (%) are as follows:
> >     Method |Comprehension & reasoning|Computation | Timeline reorder | Information retrieval| Average
> >     ---|---|----|-----|----|-----
> >     Human evaluation| 51 | 21|26|29| 36
> >     GPT4 evaluation|53| 29 | 25| 34| 38
> >     Agreement|77| 89 | 76| 80| 80
> >
> >     **It can be seen that GPT4 can function well to make human-like judgement in our dataset**. In order to provide a more comprehensive assessment, we utilize GPT4 as an LLM evaluator to obtain reference results.
> > 3) Besides, **we make our implementation reproducible and GPT4's judgment deterministic** by setting its temperature to 0, top_p to 1, and prompting GPT4 to output True/False/exact-score only, instead of descriptive results. From our observations in the experiment results, we found that the GPT4 evaluator has no bias in itself when scoring.
> > 4) **How we use GPT4-eval is also delicately designed to ensure fairness**. By giving the question(QA task only), groundtruth and predicted outputs, we ask GPT4 to compare and score considering the semantic matching, information completeness, consistency, fluency, and grammar. In this way, GPT4 can focus on the comparisons without bias and tendency for better evaluation. The detailed prompt can be seen in Appendix E.7 and E.8.
> >
> > - **For data generation**,
> > We indeed make a huge effort to avoid bias and keep fairness in evaluation:
> > 1) **For short QA**, we have manually reviewed all the short QA pairs and carefully refined the answers to make them clear and concise. The model is forced to extract the answer directly from the original document to generate the inital answer. Then we make refinements by filtering out the non-essential contexts and removing redundant descriptions from the model.
> > 2) **For long QA**, there are over 1100 long dependency QA pairs delicately designed by human annotators, despite the high costs and huge effort. Each document for generating QA pairs underwent a meticulous three-step process (Question & Answer, Answer & Check, Cross-Validation & Revise) that involved the assignment of two distinct annotators, who were unaware of each other's identities. The annotation adhered to stringent standards including long dependency, diverse problem types, clear & precise questions and deterministic & objective answers. Participants were prohibited from using large language models and tools like ChatGPT for article reading, data generation, and annotation.
> > 3) **For summarization**, answers are abstracts extracted from the paper.
> > 4) **For cloze**, we did not use gpt-3.5 to generate both the question and answer directly, and the answers are extracted by a NER model.
> >
> > **This rigorous curation process was undertaken to ensure the high quality of the questions, answers, as well as supporting evidence**. It also benefits from eliminating the interference from different model output formats, lengths, and other characteristics as much as possible.
> >
> > > Q6: In section 4.2, you mention human evaluation (3) but I cannot find any human evaluation both in the paper and the appendix.
> >
> > **A:** We have indeed included human evaluations in 3 parts of the experiments in **Appendix C**:
> > 1) We manually annotated the accuracy for 4 dependency tasks under 3 settings (without CoT, few-shot CoT, zero-shot CoT) to inspire model performance in **Figure10**.
> > 2) We provide probable explanations for long QA bad cases to provide insights and directions for model promotion in **Appendix C.1**.
> > 3) We captured discrepancies in the generated outputs of different models to tackle inherent preferences encountered in long context in **Figure 11**.
> >
> > We have also revised the paper to make this part more clearly stated for readers.

---

> > > ### Author Response · Authors · 2023-11-16
> > > **To Reviewer oc4r**
> > >
> > > > Q7: Section 4.3.2 is confusing, the experiments are based on the recent work from Liu et al. 2023, that accessing information in the middle of the document is more challenging for LLMs. If I understood correctly, you suggest concatenating the head and the tail of the inputs and give it to the LLMs as input. This would mean discarding the whole "middle" and leaving the beginning and the end. This puts an immediate limitation on information that can be retrieved by the LLMs.
> > >
> > > **A:** Thanks for your comments. Your understanding of how we truncate exceedingly long context is correct. It is the default setting in all of our experiments when the inputs are longer than LLMs' limited context window. Compared with remaining the latest inputs within the context length only, the findings in Liu et al. 2023 validates the effectiveness of enabling LLM with less information loss in the least effort.
> > >
> > >
> > > > Q8: Also did you only use the same model (GPT4-32k) but with various context input length or different variants of GPT4 ?
> > >
> > > **A:** We use the same model of GPT4-32k with varying context input length in Section 4.3.2 since limited versions of closed-source GPT4-32k and 8k can hardly support the experiment.
> > >
> > > > Q9： What is the last entry in Table 5 ? (GPT4).
> > >
> > > **A:** As we elaborated in Table 5, the last entry means GPT4-8k as a comparison. We have also revised Table 5 in the paper to make it clear.
> > >
> > >
> > > > Q10: For long summarization, arxiv abstracts are highly biased towards the beginning of the article so it is expected that increasing context would result in a higher divergence between the generated summary (which will contain more and more details) and the (gold) abstract.
> > >
> > > **A:** Thanks for your comments and thoughtful feedback. We totally agree with what you said. Due to the inherent nature of the arXiv paper, the major sketch of the paper is already included in the limited context windows. On the one hand, in our test on multiple long dependency QA tasks, the longer input indeed improves performance by introducing less information loss. On the other hand, we will continuously enrich our dataset with more data sources of various characteristics for comprehensive use and evaluation.
> > >
> > > > Q11: Typo
> > > page 2: "Mannually designed both short and long dependency tasks"
> > > page 3: "COmputation."
> > > page 7: "Retrival"
> > > page 8: "GPT4 evalution"
> > > table 4: "Performence"
> > > appendix page 3: "Dispcrepancy"
> > > not really typos but it is uninformative to report results of the order of "e-300" since at this point there is nothing to really compare.
> > >
> > > **A:** Thanks for your kind reminder and we have already fixed all the spelling errors you mentioned. Besides, we carefully walk through the whole paper and have refined the expressions and grammar issues for improvement.

---

> > > > ### Author Response · Authors · 2023-11-22
> > > > **To Reviewer oc4r**
> > > >
> > > > Dear Reviewer oc4r
> > > >
> > > > As the discussion period ends soon, we just want to reach out and see if our rebuttal answers your questions. In short, we provided:
> > > >
> > > > - dataset release and implementation details
> > > > - clarification of GPT4 & human evaluation, way of truncation
> > > > - additional experiments on instruct-tuned open source models for comparison
> > > >
> > > > Thank you again for your comments and suggestions to improve our paper! We look forward to your reply.

---

### Official Review · Reviewer_SWcJ · 2023-11-01

**Soundness:** 2 fair
**Presentation:** 1 poor
**Contribution:** 2 fair
**Rating:** 3
**Confidence:** 5

**Summary:**

LLMs have shown impressive performance in various NLP tasks. However, the fixed context window length of the transformer architecture limits their ability to understand extremely long inputs. Existing datasets for evaluating LLMs' long context understanding have limitations such as shorter text lengths, outdated documents, and a focus on short dependency tasks. The paper introduces "LooGLE"  to evaluate LLMs' ability to understand long contexts. Upon evaluating 8 state-of-the-art LLMs on LooGLE, the authors found:
1. Commercial models generally outperform open-sourced models.
2. LLMs excel at short dependency tasks but struggle with real long dependency tasks.
3. Retrieval-based techniques significantly improve performance on short QA tasks, but many techniques for extending context window length struggle with long context understanding.

**Strengths:**

- **Up-to-date Documents**: LooGLE contains documents published after 2022, ensuring that modern LLMs have not been pretrained on these documents.
- **Diverse Tasks**: LooGLE includes both short and long dependency tasks, providing a evaluation of 8 LLMs' capabilities.

**Weaknesses:**

**Lack of Experimental Data to Support Some Claims**:

- The paper states that "by employing scaling techniques like positional interpolation, parallelization, and finetuning on longer texts, open-sourced models have shown improvement in handling longer inputs compared to previous versions." However, the article does not provide performance data of previous version models. This omission makes it challenging to ascertain the improvements is brought by modifying position embeddings or instruction-tuning. For example, the comparision between vicuna-2k and vicuna-16k is a better case to validate this claim.
- The claim that "GPT4-32k performs better than GPT-8k" is not consistently supported by the provided metrics. The results between the two models vary across different indicators (automatic metrics v.s. gpt4 score). A more in-depth explanation and analysis are needed to support this claim, including understanding the differences in various metrics. It's unclear why, on Long dependency tasks, the longer window 32k model performs worse than the 8k model.

**Absence of Human Evaluation**: The paper mentions conducting human evaluations, but there's no presentation of the related data. GPT-evals may have some preference in generation length, human evaluation can be a better reference.

**Questions:**

- Details of Llamaindex is not clear: 7B or 13B, chat model or regular model?
- Why not use the chat version of Llama2, which is considered to be skilled at instruction-following and could be potentially better at downstream tasks.
- Why LlamaIndex is much better than any other open-sourced models? According to your results in Fig3, retrieval+open-sourced model is better than the long-context version of the same base model, and this conclusion is contradict to the conclusion from other long-context benchmarks (Table1).
- Writing format: it's better to list url as the footnote. It is supposed to leave a black between the main text and citation brackets.

---

> ### Author Response · Authors · 2023-11-16
> **To Reviewer SWcJ**
>
> > Q1: The paper states that "by employing scaling techniques like positional interpolation, parallelization, and finetuning on longer texts, open-sourced models have shown improvement in handling longer inputs compared to previous versions." However, the article does not provide performance data of previous version models. This omission makes it challenging to ascertain the improvements is brought by modifying position embeddings or instruction-tuning. For example, the comparision between vicuna-2k and vicuna-16k is a better case to validate this claim.
>
> **A:** Thanks the reviewer for your constructive advice.
> - We have to emphasize that the paper only claims "**open-sourced models have shown improvement in handling longer inputs** by employing the scaling techniques" instead of gaining performance improvements in downstream tasks.
> - However, we also **agree with your suggestion and indeed explore this by adding additional experiments**. We further compare the varying input lengths of ChatGLM2-6B-32k (32k context window) with ChatGLM2-6B (8k context window) on long dependency QA. We select ChatGLM2 as the testing model since it achieves better performance in our previous experiments among different open-sourced models. Here are some findings:
>     1) The original version model ChatGLM2-6B performs better than ChatGLM2-6B-32k across the automated metrics given the same length of inputs. There is a performance decline for the model when extending the longer context window. It can be attributed to the information loss introduced by the scaling techniques of long context models, which calls for further improvement on open-sourced LLMs.
>     2) Moreover, for ChatGLM2-6B-32k with varying lengths of inputs, we find that the extension of longer inputs has limited impact on performance. It is because the long dependency QAs in our dataset request for the long dependency comprehension and  modeling capability, along with further computation and reasoning. Our dataset propose higher demands on true long context undetstanding, which needs to be desperately resolved and enhanced for further LLMs.
>
> | Models         | Context | Bleu1 | Bleu4 | Rouge1 | Rouge4 | RougeL | Meteor_score | Bert_score | GPT4 evaluator |
> |----------------|---------|-------|-------|--------|--------|--------|--------------|------------|----------------|
> | ChatGLM-6b-32k | 32k     | 5.62  | 0.01  | 11.95  | 1.45   | 10.84  | 5.55         | 87.18      | 20.64          |
> | ChatGLM-6b-32k | 24k     | 7.04  | 0.16  | 13.74  | 2.67   | 12.80  | 6.10         | 87.93      | 20.00          |
> | ChatGLM-6b-32k | 16k     | 6.37  | 0.01  | 13.26  | 1.49   | 12.29  | 5.69         | 87.87      | **21.53**          |
> | ChatGLM-6b-32k | 8k      | 4.98  | 0.11  | 11.56  | 1.68   | 10.79  | 5.26         | 87.90      | 21.37          |
> | ChatGLM-6b     | 8k      | **9.72**  | **0.43**  | **14.03**  | **2.79**   | **13.12**  | **9.53**         | **88.74**      | 20.11          |

---

> > ### Author Response · Authors · 2023-11-16
> > **To Reviewer SWcJ**
> >
> > > Q2：The claim that "GPT4-32k performs better than GPT-8k" is not consistently supported by the provided metrics. The results between the two models vary across different indicators (automatic metrics v.s. gpt4 score). A more in-depth explanation and analysis are needed to support this claim, including understanding the differences in various metrics. It's unclear why, on Long dependency tasks, the longer window 32k model performs worse than the 8k model.
> >
> > **A:** Thanks for pointing out. We summarize the performance for both models as below:
> > - **For summarization evaluated by both automatic metrics and GPT4-eval**, GPT4-8k performs better where longer context window has marginal improvement from Table 4 and Table 5. It is due to the intrinsic features of ArXiv papers with both the introduction and conclusion located at the beginning and in the end respectively. In this way, GPT4-8k with limited window size can still performs well to contain the major sketch of the paper.
> > - **For cloze, short & long QA evaluated by exact/partial match and GPT4-eval**,  GPT4-32k performs better semantically with more complete inputs with less information loss and stronger context understanding capability from Table 3, 4, 5.
> > - **For QA tasks evaluated by other automatic metrics**, GPT4-32k performs worse in some cases from Table 3. It can be attributed to its tendency to generate longer outputs, faces penalties from these automatic metrics. It is more evident in short dependency QA where GPT4-8k can extract the right answer with limited window.
> > - The disagreement between metrics lies in the reason that each of them **measures the generated texts from different aspects**. For instance, BLEU and ROUGE evaluate n-gram based similarity but compute precision and recall respectively. METEOR incorporates matching in various aspects, including word stemming and variations, syntax, order, and phrase structure, and computes a semantic matching score. BERTScore leverages pre-trained contextual embeddings and computes the cosine similarity from token level. Exact Match entails precise comparison between the predicted entities and the ground truth entities while Partial Match allows for fuzzy matching. Therefore, we use various automatic metrics commonly used to obtain comprehensive performance evaluations.
> >
> > Therefore, we use various evaluation methods in our paper to obtain comprehensive and fair performance evaluations. We have also reviesed the corresponding expressions in the paper to make it clearer.
> >
> >
> > > Q3：The paper mentions conducting human evaluations, but there's no presentation of the related data. GPT-evals may have some preference in generation length, human evaluation can be a better reference.
> >
> > **A:** We have indeed inclued human evaluations in 3 parts of the experiments in **Appendix C**:
> > 1) We manually annotated the accuracy for 4 dependency tasks under 3 settings (without CoT, few-shot CoT, zero-shot CoT) to inspire model performance in **Figure10**.
> > 2) We provide probable explanations for long QA bad cases to provide insights and directions for model promotion in **Appendix C.1**.
> > 3) We captured discrepancies in the generated outputs of different models to tackle inherent preferences encountered in long context in **Figure 11**.
> >
> > We have also revised the paper to make this part more clearly stated for readers.

---

> ### Author Response · Authors · 2023-11-16
> **To Reviewer SWcJ**
>
> > Q4: Details of Llamaindex is not clear: 7B or 13B, chat model or regular model?
>
> **A:** We use the official default model text-davinci-003 for LlamaIndex in all the experiments.
>
>
> > Q5: Why not use the chat version of Llama2, which is considered to be skilled at instruction-following and could be potentially better at downstream tasks.
>
> **A:** We thank the reviewers for your useful comments.  To the best of our knowledge, LLaMA2-7B-32K used in this paper is fine-tuned from Meta's original Llama-2 7B model to extend context and also instruction-tuned for summarization and long context QA for better performance. We have highlighted this information in Section 4.1 in the revised paper.  We are happy to extend to new models and will keep updating the leaderboard.
>
> > Q6: Why LlamaIndex is much better than any other open-sourced models? According to your results in Fig3, retrieval+open-sourced model is better than the long-context version of the same base model, and this conclusion is contradict to the conclusion from other long-context benchmarks (Table 1).
>
> **A:** Thank you for the comments.
> - LlamaIndex is a retrieval-based data pipeline for different models. We use text-davinci-003 as default model in our experiment instead of the open-source Meta Llama.
> - From the short QA in Table 3, retrieval-based techniques indeed demonstrate their benefits for short QA. However, they have a limited impact on understanding long context for long dependency tasks, as shown in Table 4.
> - Our conclusion in Table 1 is consistent with that in LongBench. However, we have quite a different experiment settings (designed tasks, evaluation metrics, etc.) to drive to this conclusion.
>
>
> > Q7: Writing format: it's better to list url as the footnote. It is supposed to leave a black between the main text and citation brackets.
>
> **A:** Thanks for your advice and we have already moved all the urls as the footnotes the writing format according to your comments (see Page 4 and 7). Also, we have refined the writing format issue of blank you mentioned.

---

> > ### Author Response · Authors · 2023-11-22
> > **To Reviewer SWcJ**
> >
> > Dear Reviewer SWcJ
> >
> > As the discussion period ends soon, we just want to reach out and see if our rebuttal answers your questions. In short, we provided:
> >
> > - additional experiments to support the claims you concerned
> > - clarification of existing human evaluation
> > - in-depth explanations on the comparison between GPT4-32k and GPT4-8k,  performance on llamaindex
> > - implementation details on baseline models eg. llama2, llamaindex
> >
> > Thank you again for your comments and suggestions to improve our paper! We look forward to your reply.

---

### Official Review · Reviewer_gE1R · 2023-11-06

**Soundness:** 4 excellent
**Presentation:** 3 good
**Contribution:** 4 excellent
**Rating:** 8
**Confidence:** 4

**Summary:**

The authors present a new dataset, called LooGLE, which aims at evaluation of LLMs on long context. Their dataset has documents with longer length compared to previous benchmarks and it is more up to date (2022+). The proposed dataset has task with long dependencies and the authors have made sure that for some of the tasks, the answer needs to be collected from multiple segments of the documents. The evaluate both commercial and open-souse models on the new dataset and provide some insights.

**Strengths:**

- The paper addresses a very important area, i.e., long context evaluation of LLMs
- Based on the description, the collection method, and evaluation results, the proposed new dataset seems to be of high quality.
- Authors provide extensive evaluation on different commercial and open-sourced models.
- The paper is well written and easy to follow.

**Weaknesses:**

- The paper presents Human Evaluations as one of the evaluation technique but never present human evaluation results.
- There are several automatic scores have been presented in Tables 3 through 5 and sometimes. These scores not always in agreement; not all the scores are better for the winning model. I found this confusing especially when some conclusions are drawn in the text.

**Questions:**

- For the LlamaIndex, what retriever and chunk size are used?

---

> ### Author Response · Authors · 2023-11-16
> **To Reviewer gE1R**
>
> We sincerely thank the reviewer for your acknowledgment and constructive comments on our work. We would like to involve further discussions in the following sections.
>
>
>  > Q1: The paper presents Human Evaluations as one of the evaluation technique but never present human evaluation results.
>
> **A:** We have indeed included human evaluations in 3 parts of the experiments in **Appendix C**:
> 1) We manually evaluated the accuracy for 4 long-dependency tasks under 3 settings (without CoT, few-shot CoT, zero-shot CoT) in **Figure 10**.
> 2) We provide probable explanations for long QA bad cases to provide insights and directions for model promotion in **Appendix C.1**.
> 3) We captured discrepancies in the generated outputs of different models to tackle inherent preferences encountered in long context in **Figure 11**.
>
> We have also revised the paper to make this part more clearly stated for readers.
>
>
> > Q2: There are several automatic scores have been presented in Tables 3 through 5 and sometimes. These scores not always in agreement; not all the scores are better for the winning model. I found this confusing especially when some conclusions are drawn in the text.
>
> **A:** Thanks for pointing out. We summarize the main performance in the experiments and provide potential explanations  as below:
> - **For summarization evaluated by both automatic metrics and GPT4-eval**, GPT4-8k performs better while longer context window has marginal improvement from Table 4 and Table 5. It is due to the intrinsic features of ArXiv papers with both the introduction and conclusion located at the beginning and in the end respectively. In this way, GPT4-8k with limited window size can still perform well to capture the major sketch of the paper.
> - **For cloze, short & long QA evaluated by exact/partial match and GPT4-eval**,  GPT4-32k performs better semantically with more complete inputs with less information loss and stronger context understanding capability from Table 3, 4, 5.
> - **For QA tasks evaluated by other automatic metrics**, GPT4-32k performs worse in some cases from Table 3. It can be attributed to its tendency to generate longer outputs, faces penalties from these automatic metrics. It is more evident in short dependency QA where GPT4-8k can extract the right answer with limited window.
>
> The disagreement between metrics lies in the reason that each of them **measures the generated texts from different aspects**. For instance, BLEU and ROUGE evaluate n-gram based similarity but compute precision and recall respectively. METEOR incorporates matching in various aspects, including word stemming and variations, syntax, order, and phrase structure, and computes a semantic matching score. BERTScore leverages pre-trained contextual embeddings and computes the cosine similarity from token level. Exact Match entails precise comparison between the predicted entities and the ground truth entities while Partial Match allows for fuzzy matching. Therefore, we use various automatic metrics commonly to obtain comprehensive performance evaluations.
>
> We have also revised the corresponding expressions in the paper to make it clear.
>
> > Q3: For the LlamaIndex, what retriever and chunk size are used?
>
> **A:** We use the official default model text-davinci-003 as the retriever and default chunk size (1024) for LlamaIndex.

---

### Author Response · Authors · 2023-11-16
**General Response**

We sincerely thank all reviewers for their valuable comments and constructive suggestions! The main contributions of our work and the important questions are highlighted and answered as follows:

**1. Main strengths and innovation**
- **Contributory work in the important area.** We focus on the "long context evaluation and improvement of LLMs" (Reviewer **gE1R**), which is "usefully needed and contributes to the community" (Reviewer **8ZV2**). LooGLE sheds light on future development of enhanced models towards “true long-context understanding”. It shows current LLMs targeting on long-context understanding all fail to solve the true long-dependency problems, and merely increasing the context window size might not help.
- **Novel dataset for long context.** The dataset has the remarkable advantages of extra-long realistic documents with over 24k tokens on average, "diverse tasks with manually designed both short and long dependency tasks, up-to-date documents (all after 2022) to prevent data leakage" (Reviewer **SWcJ**) and 6k newly generated questions spanning diverse domains and categories.
- **Curated dataset thoughtfully designed of high quality** LooGLE is "a curated dataset, thoughtfully designed with great efforts to prevent data leakage and ensure long dependency" (Reviewer **oc4r**, **8ZV2**). We spend huge effort and cost throughout the whole process of "data collection & selection, task generation and cross-validation as well as model evaluation to guarantee the high quality of the dataset" (Reviewer **gE1R**).
- **Comprehensive Experiments.** Our work "provides extensive evaluations on different commercial and open-sourced long context models" (Reviewer **gE1R**) under three evaluation methods and various metrics. We spent "great efforts assessing the current state-of-the-art LLMs' long dependency capabilities, such as information retrieval, reading comprehension and reasoning, computation and timeline reordering" (Reviewer **oc4r**).

**2. Data release and applicability**

Currently, our polished dataset is **released in the HuggingFace Datasets** and can be easily downloaded anytime for use. But to **meet the anonymity requirement from the conference**, we couldn’t make it public in this paper and hope for your understanding. The current version of our dataset is English-only and provides all the examples as testing set for public. Meanwhile, we also provide detailed instructions on the acquiration and usage of the dataset in the main paper of LooGLE as well as github for better use and dataset reproducibility. We will contiously and periodically update the dataset with newer documents, diverse and challenging tasks, as well as assessment of latest long context LLMs possibly on a yearly basis. Besides, our data collection process is fully open-sourced, making it easy for the community to reconstruct/update the benchmark.

We've revised our manuscript per the reviewers' suggestion (highlighted in red in the uploaded revision pdf). Detailed responses to each reviewer's concerns are carefully addressed point-by-point. Below we summarize the major updates we've made:
- Delicately reorganize the results and clarify the conclusions for the main evaluations to make the results clearer. (**Section 4.3**)
- Meticulously refine the expressions and logic for results of deep analysis on long context capabilities. (**Appendix C**)
- Carefully refine the presentation of the paper including main figures and tables. (**Figure 1 and 3, Table 1 and 2**)